# Sustainability characteristics of drinking water supply in the Netherlands

*Jolijn van Engelenburg[1], Erik van Slobbe[2], Adriaan J. Teuling[3], Remko Uijlenhoet[3,4], Petra Hellegers[5]*

[1] Asset Management Department, Vitens NV, P.O.Box 1205, 8001 BE Zwolle, Netherlands

[2] Water Systems and Global Change Group, Wageningen University & Research, P.O. Box 47, 6700 AA Wageningen, Netherlands

[3] Hydrology and Quantitative Water Management Group, Wageningen University & Research, P.O. Box 47, 6700 AA Wageningen, Netherlands

[4] Water Management Department, Civil Engineering and Geosciences Faculty, Delft University of Technology P.O. Box 5048, 2600 GA Delft, Netherlands

[5] Water Resources Management Group, Wageningen University & Research, P.O. Box 47, 6700 AA Wageningen, Netherlands

**Corresponding author:**

Jolijn van Engelenburg

Vitens NV

P.O. Box 1205, 8001 BE Zwolle

Netherlands

Telephone: +31651525936

E-mail: jolijn.vanengelenburg@wur.nl; jolijn.vanengelenburg@vitens.nl; jvanengelenburg247@gmail.com

**Biography corresponding author**

Asset manager at Drinking water company Vitens, the Netherlands

**Keywords**

Systems approach; DPSIR; drinking water supply; local scale; sustainability

## Abstract

Developments such as climate change and growing demand for drinking water threaten the sustainability of drinking water supply worldwide. To deal with this threat, adaptation of drinking water supply systems is imperative, not only on a global and national scale, but particularly on a local scale. This investigation sought to establish characteristics that describe the sustainability of local drinking water supply. The hypothesis of this research was that sustainability characteristics depend on the context that is analysed and therefore a variety of cases must be analysed to reach a better understanding of the sustainability of drinking water supply in the Netherlands. Therefore three divergent cases on drinking water supply in the Netherlands were analysed. One case related to a short-term development (2018 summer drought), and two concerned long-term phenomena (changes in water quality and growth in drinking water demand). We used an integrated systems approach, describing the local drinking water supply system in terms of hydrological, technical and socio-economic characteristics that determine the sustainability of a local drinking water supply system. To gain a perspective on the case study findings broader than the Dutch context, the sustainability aspects identified were paired with global aspects concerning sustainable drinking water supply. This resulted in the following set of hydrological, technical and socio-economic sustainability characteristics: (1) water quality, water resource availability, and impact of drinking water abstraction; (2) reliability and resilience of the technical system, and energy use and environmental impact; (3) drinking water availability, water governance, and land and water use. Elaboration of these sustainability characteristics and criteria into a sustainability assessment can provide information on the challenges and trade-offs inherent in the sustainable development and management of a local drinking water supply system.

# 1 Introduction

Climate change combined with a growing drinking water demand threatens the sustainability of the drinking water supply worldwide. The goal set for drinking water supply in Sustainable Development Goal (SDG) 6.1 (UN, 2015) is "to achieve universal and equitable access to safe and affordable drinking water for all by 2030". Worldwide drinking water supply crises are visible, resulting from a combination of limited water resource availability, lacking or failing drinking water infrastructure and/or increased drinking water demand, due to short-term events or long-term developments (WHO, 2017b). Still, nearly 10 percent of the world population is fully deprived of improved drinking water resources (Ekins et al., 2019), and, additionally, existing drinking water supply systems often are under pressure. For instance, two recent examples of water crises were reported in Cape Town, South Africa and São Paolo, Brazil (Sorensen, 2017, Cohen, 2016). To deal with such challenges and threats to safe and affordable drinking water, adaptation of the current drinking water supply system is imperative, not only on a global and national level, but also on a local scale.

In the Netherlands, for instance, the national drinking water supply currently meets the indicator from SDG 6 (UN, 2018) on safely managed drinking water services and safely treated waste water. At the same time the more specific goals on (local) water quantity, quality, and ecology as set by the European Water Framework Directive (WFD), are not met yet (European Environment Agency, 2018). Consequently, drinking water supply in the Netherlands does not meet all SDG 6 indicators, for instance when considering impact to water-related ecosystems (Van Engelenburg et al., 2017), water pollution (Kools et al., 2019, Van den Brink and Wuijts, 2016), or water shortage (Ministry of Infrastructure and Environment and Ministry of Economic Affairs and Climate Policy, 2019). Additionally, future developments such as the

uncertain drinking water demand growth rate (Van der Aa et al., 2015) and the changing
climate variability (Teuling, 2018), may put the sustainability of the Dutch drinking water
supply under pressure in the future.
The abstraction of groundwater or surface water from the hydrological system, and
subsequent treatment to drinking water quality before being distributed to customers,
requires a local infrastructure (typically a drinking water production facility, embedded in a
distribution network of pipelines). Although the daily routine of drinking water supply has a
highly technical character (Bauer and Herder, 2009), the sustainability in the long-term
depends on the balance between technical, socio-economic and environmental factors. This
balance is especially complex for local drinking water supply, which is intertwined with the
local hydrological system and local stakeholders through its geographical location.
Because of the interconnections between physical, technical, and socio-economic factors as
well as across space, organizational levels and time, adaptation of the local drinking water
supply to current and future sustainability challenges calls for an integrated planning approach
(Liu et al., 2015). Integrated models have been developed to understand the complex
interactions between the physical, technical and socio-economic components in various water
systems (Loucks et al., 2017). However, a systems analysis to assess local drinking water supply
and to identify sustainability challenges on a local scale has not yet been developed.
This research aimed to propose a set of sustainability characteristics that describe the drinking
water supply system on a local scale to support policy- and decision-making on sustainable
drinking water supply. To reach this aim, cases on drinking water supply were analysed using
a conceptual framework. The selected cases represented a short-term event and long-term
developments that affect water quality and water resource availability, the technical drinking

water supply infrastructure and/or the drinking water demand. The system boundaries were set to drinking water supply on the local scale. While the drinking water supply on a local scale is also affected by outside influences from different organizational and spatial scales, the analysis accounted for these external influences too. The hypothesis of this research was that sustainability characteristics depend on the context that is analysed and therefore a variety of cases must be analysed to reach a better understanding of the sustainability of drinking water supply in the Netherlands.

## 2   Method

Sustainable water systems can be defined as water systems that are designed and managed to contribute to the current and future objectives of society, maintaining their ecological, environmental, and hydrological integrity (Loucks, 2000). This study focused on the sustainability of drinking water supply systems on a local scale, in short, local drinking water supply systems. The boundaries of these systems were set by the area in which drinking water abstraction is embedded. The system can be approached from different perspectives. The socio-ecological approach considers relations between the socio-economic and environmental system, whereas the socio-technical approach considers the socio-economic and technical system (Pant et al., 2015). In this study we combined both approaches by describing the local drinking water supply system in terms of hydrological, technical and socio-economic characteristics that determine the sustainability of a local drinking water supply system.

Drinking water supply in the Netherlands is of a high standard compared to many other countries. The SDG 6 targets on safe and affordable drinking water and sanitation and

wastewater treatment are basically met. But the Dutch government and drinking water
suppliers are also challenged to meet the other goals set in SDG 6, such as improvement of
water quality, increase of water-use efficiency, and protection and restoration of water-
related ecosystems. In addition the standards on water quantity, quality, and ecology as set
by the European Water Framework Directive (WFD) have not been achieved yet (European
Environment Agency, 2018).
The adopted research approach consisted of four steps. The first step was the selection and
analysis of three drinking water practice cases in the Netherlands, aiming to identify the main
Dutch sustainability aspects in these cases. Three Dutch cases were selected based on their
impact to the sustainability of drinking water supply in the Netherlands, considering a short-
term event with limited water resource availability, as well as long-term ongoing
developments on water quality, and growing drinking water demand and water resource
availability. The cases are illustrated with Vitens data (Van Engelenburg et al., 2020b).
In the second step the cases were analysed using the DPSIR framework (*Driver, Pressure, State,*
*Impact, Response* (Eurostat, 1999), see section 2.1). The sustainability aspects of these cases
were identified in the descriptive results of the DPSIR analysis. The results were combined
with Dutch governmental reports on these events and developments (Ministry of
Infrastructure and Environment and Ministry of Economic Affairs and Climate Policy, 2019,
Vitens, 2016) and cross-checked with Vitens staff. The sustainability aspects were categorized
into hydrological, technical and socio-economic aspects. This resulted in a set of relevant
sustainability aspects, which is presented in Appendices A-C. The following step was used to
broaden the perspective from the drinking water supply in the Netherlands to a more general
perspective, by cross-checking the set of sustainability aspects with the targets and indicators

in Sustainable Development Goal 6 (further referred to as "SDG 6", see App. D) (UN, 2015), and the WHO Guidelines for Drinking-Water Quality (WHO, 2017a). The *sustainability aspects* as identified in the analysis were categorized into nine hydrological, technical and socio-economic *sustainability characteristics*. In the final step of the study each sustainability characteristic was elaborated further into five sustainability criteria that describe the local drinking water supply system. The results are described in section 3. A detailed description of the resulting sustainability criteria is presented in Appendix E.

## 2.1 Case analysis method

To reach the aim of this research to support policy development on sustainable drinking water supply, three practice cases were analysed to identify the main sustainability aspects in these cases using the DPSIR (*Driver-Pressure-State-Impact-Response*) systems approach (Eurostat, 1999): *Drivers* describe future developments such as climate change and population growth. *Pressures* are developments (in emissions or environmental resources) as a result from the drivers. The *state* describes the system state that results from the pressures. In this research the aim is to describe the system state of the drinking water supply system in terms of local hydrological, technical and socio-economic sustainability characteristics (see section 2.1). The changes in system state cause *impacts* to system functions, which will lead to societal *responses*. DPSIR was originally developed to describe causal relations between human actions and the environment. It has also frequently been used for relations and interactions between technical infrastructure and the socio-economic and physical domain (Pahl-Wostl, 2015, Hellegers and Leflaive, 2015, Binder et al., 2013).

The DPSIR approach was used for the analysis of the three selected drinking water supply cases to obtain an overview of the *impact* of *drivers, pressures* and *responses* to the *state* of

the drinking water supply system. Although the framework has been applied on different
spatial scales, Carr et al. (2009) recommend using the framework place-specific, to ensure that
local stakeholder perspectives are assessed as well. With the research focus at the local
drinking water supply system, these local perspectives were implicitly included. The *drivers*,
*pressures* and *responses* can be on local as well as higher organizational and/or spatial scales,
thus ensuring that - where essential - relevant higher scales are accounted for too.
DPSIR has previously been used for complex water systems by various well-known
researchers in this field, such as Claudia Pahl-Wostl. In Binder et al. (2013) a comparison was
made between various frameworks, which concluded that DPSIR is a policy framework that
does not explicitly include development of a model, but aims at providing policy relevant
information, on pressures and responses on different scales. In Carr et al. (2009) the use of
DPSIR for sustainable development was evaluated. Although the authors were critical
regarding the use of the DPSIR framework on national, regional or global scales, they
considered application on a local scale appropriate. They concluded that practitioners can
use DPSIR for local-scale studies because it assesses the place-specific nuances of multiple
concerned stakeholders more realistically. In Van Noordwijk et al. (2020) DPSIR was used to
understand the joint multi-scale phenomena in the forest-water-people nexus and thus
diagnosed issues to be addressed in serious games for local decision-making. Therefore,
DPSIR was considered an appropriate framework to meet the aim of the research.
The impact of developments on different temporal scales to the drinking water supply system
must be considered as well. The long lived, interdependent drinking water supply
infrastructure is rigid to change due to design decisions in the past, which is causing path-
dependencies and lock-ins (Melese et al., 2015). In addition, consumer behaviour, governance
and engineering, and the interaction between these processes cause lock-in situations that
limit the ability to change towards more sustainable water resources management (Pahl-
Wostl, 2002). For this reason, the case analysis was performed considering both short- and
long-term *pressures*, *impacts* and *responses*.

## 2.2  Case selection

In this research three drinking water supply cases in the Netherlands were selected. The case
studies were analysed to find sustainability aspects caused by the identified *pressures* and
short- and/or long-term *responses* in each case, because short-term shocks have different
*impacts* and call for other *responses* than long-term stresses (Smith and Stirling, 2010). The
cases therefore focused on short-term events as well as long-term developments. All three
cases also related to targets set in SDG 6 (UN, 2015). The DPSIR analysis of the case studies is
presented in Appendices A-C.

**Case 1 "2018 Summer drought"**

Summer 2018 in the Netherlands was extremely warm and dry, causing water shortages in the
water system, and a long period of extreme daily drinking water demand, resulting in a record
monthly water demand in July 2018 (Ministry of Infrastructure and Environment and Ministry
of Economic Affairs and Climate Policy, 2019) (see Illustration case 1). The *driver* in this case is
the extreme weather condition, which caused several *pressures*, such as high temperatures,
high evaporation and lack of precipitation. These *pressures* did not only cause drought damage
to nature, agriculture and gardens and parks, as well as limited water availability in the surface
water and groundwater systems, they also resulted in an extremely high drinking water
demand. Data on drinking water supply volumes (Van Engelenburg et al., 2020b) showed that
the extreme drinking water demand during summer 2018 put the drinking water supply
system under high pressure, causing extreme daily and monthly drinking water supply
volumes that exceeded all previously supplied volumes (see Fig. 1). The capacity of the system
was fully exploited, but faced limitations in abstraction, treatment and distribution capacity.

*Illustration case 1: 2018 Summer drought*

*Within the Vitens supply area the average daily supply volume during the summer period June-August over the years 2012-2017 was approximately 965,000 m$^3$/day. During the period 27 June-4 August 2018 the daily supply volume exceeded this average summer volume with approximately 28%, with an average volume of nearly 1,240,000 m$^3$/day (Fig. 1a). On 25 July 2019 the maximum daily water supply reached nearly 1,390,000 m$^3$/day, which was 42% above the baseline daily supply (Fig. 1a). The monthly drinking water supply volume in July 2018 of 38 million m$^3$/month was an increase of 18% compared to the previous maximum monthly supply volumes (Fig. 1b). Although the drinking water supply infrastructure was designed with an overcapacity to meet the regular demand peaks, the flexibility to more extreme peaks, or to long periods of peak demand is limited.*

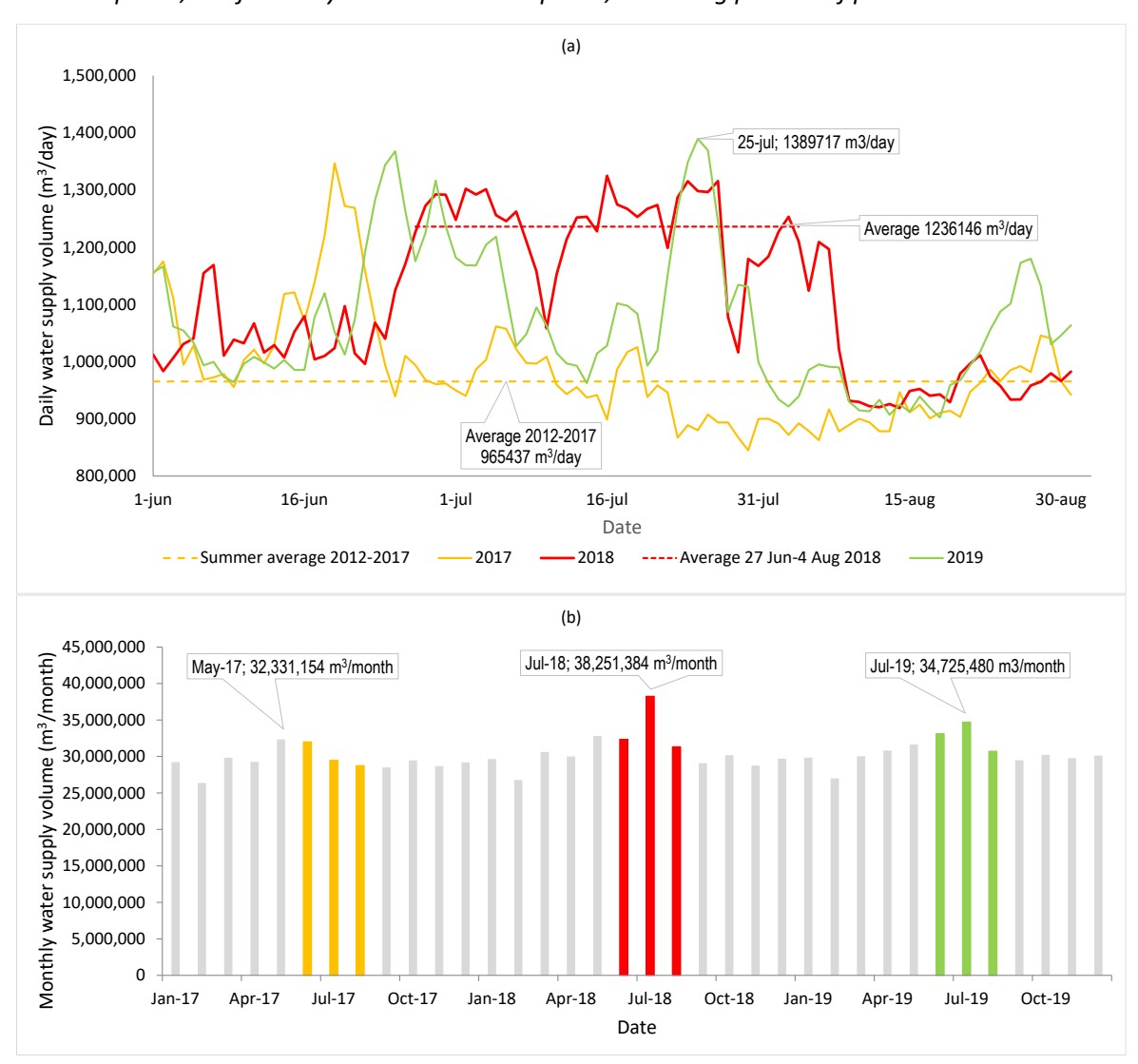

**Figure 1** *Daily (a) and monthly (b) drinking water supply volume by Dutch drinking water supplier Vitens during summer 2017 (average), 2018 (extreme), 2019 (high) (Van Engelenburg et al., 2020b).*

The high drinking water abstraction volumes added up to the water shortages in both the
groundwater and the surface water system caused by the lack of precipitation and high
evaporation during the summer (Ministry of Infrastructure and Environment and Ministry of
Economic Affairs and Climate Policy, 2019). To ensure an acceptable surface water quality for
the drinking water supply, measures were taken to reduce salinization (Ministry of
Infrastructure and Environment and Ministry of Economic Affairs and Climate Policy, 2019).
To reduce the drinking water use, a call for drinking water saving was made, and locally
pressures in the drinking water distribution system were intentionally lowered to reduce the
delivered drinking water volumes (Ministry of Infrastructure and Environment and Ministry of
Economic Affairs and Climate Policy, 2019). The problems caused by the summer drought
raised a discourse on (drinking) water use and saving, including discussions on controversial
measures such as a progressive drinking water tariffs, with tariffs dependent on the consumed
drinking water volume, and differentiation between high-grade and low-grade use of
(drinking) water (Ministry of Infrastructure and Environment and Ministry of Economic Affairs
and Climate Policy, 2019).  The results of this case analysis are presented in App. A.
**Case 2 "Groundwater quality development"**
This case focused on the impact of the groundwater quality development in the Netherlands
to the drinking water supply. Analysis of the state of the resources for drinking water supply
in the Netherlands in 2014 pointed out that, although the drinking water quality met the Dutch
legal standards, all water resources are under threat by known and new pollutants (Kools et
al., 2019). In the Netherlands 55% of the drinking water supply is provided by groundwater
resources (Baggelaar and Geudens, 2017). Long-term analysis of water quality records of
Dutch drinking water supply fields shows that the vulnerability of groundwater resources to

external influences such as land use strongly depends on hydrochemical characteristics (Mendizabal et al., 2012). Monitoring results show that currently groundwater quality is mainly under pressure due to nitrate, pesticides, historical contamination and salinization (Kools et al., 2019). Nearly half of the groundwater abstractions for drinking water are affected by an insufficient groundwater quality, and it is expected that in the future the groundwater quality at more abstractions will exceed the groundwater standards set in the European Water Framework Directive (European Union, 2000). In addition, traces of pollutants such as recent industrial contaminants, medicine residues and other emerging substances are found, indicating that the groundwater quality will likely further deteriorate (Kools et al., 2019).

Groundwater protection regulations regarding land and water use by legal authorities will help to slow down groundwater deterioration (Van den Brink and Wuijts, 2016). However, strategies to restore groundwater quality often will not be effective in the short term, because already existing contaminations may remain present for a long period of time, depending on the local hydrological characteristics (Jørgensen and Stockmarr, 2009) (see Illustration case 2). The impact of contamination cannot be undone, unless soil processes help to (partially) break down contaminants. Thorough monitoring for pollution therefore is essential to follow groundwater quality trends and to respond adequately to these trends (Janža, 2015). Due to the expected deterioration of the raw water quality[1], different and more complex treatment methods are necessary to continuously meet the drinking water standards (Kools et al., 2019). In general, a more complex treatment method leads to higher energy use, use of additional excipients, water loss and production of waste materials, which

---

[1] Raw water is the (untreated) water that is treated to produce drinking water. This can be abstracted groundwater or surface water depending on the available water resource.

will lead to a higher water tariff, and to a higher environmental impact (Napoli and Garcia-
Tellez, 2016). The results of the analysis are presented in App. B.

---

*Illustration case 2: Groundwater quality development*

*In the 1980's the Dutch government installed regulations to protect water quality by limiting the growing nitrate and phosphate surplus due to overuse of livestock manure. This resulted in a decrease of the nitrate surplus from 1985 on. However, due to the long travel times in groundwater it took years before the impact of these regulations became visible in the groundwater quality. Fig. 2 illustrates the period of time in which the nitrate concentration in an abstraction well still increased despite the 1985 regulations on reduction of the nitrate surplus at surface level: the nitrate concentration in this well has increased until 2005 before the nitrate level started to decrease. Only since 2014 the concentration has dropped below the nitrate standard for groundwater of 50 mg/L.*

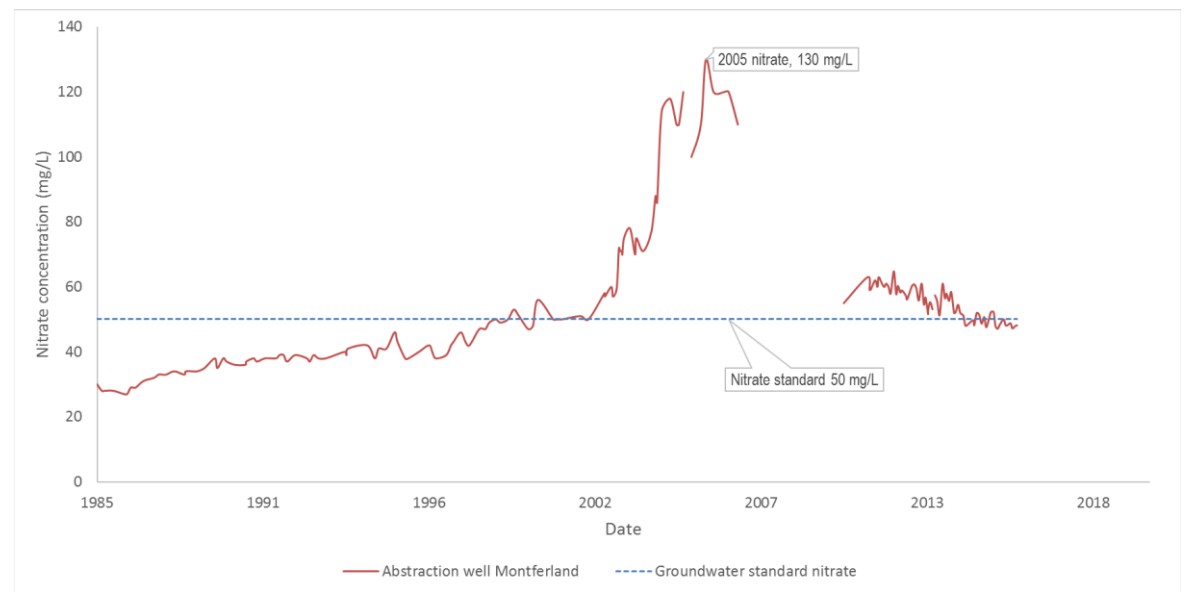

**Figure 2** *Development of nitrate in an abstraction well in Montferland (HEE-P07-07.0, coordinates X213.540-Y434.761) in the province of Gelderland, the Netherlands (Van Engelenburg et al., 2020b) compared to the Dutch standard for nitrate concentration in groundwater (50 mg/L).*


---

**Case 3 "Drinking water demand growth"**
Due to drinking water saving strategies the drinking water use in the Netherlands per person
has decreased from 137 litre per person per day in 1992 to 119 litre per person per day in
2016 (Van Thiel, 2017). This development resulted in a decreasing total yearly drinking water
demand volume in that same period, despite the population growth in the Netherlands
(Baggelaar and Geudens, 2017). However, 2013 was a turning point, when the total yearly
drinking water demand volume in the Netherlands started to grow again (Baggelaar and
Geudens, 2017). The trend in the period 2013-2019 shows a strong increase in drinking water
demand (see Illustration case 3). Delta scenarios have been developed for the Netherlands,
projecting a drinking water demand development varying between a decrease of 10% to an
increase of 35% in 2050 compared to 2015 (Wolters et al., 2018).
The drinking water demand growth rate of the period 2013-2019 as is seen within the Vitens
supply area compares to the growth rate in the maximum delta scenario of 35% growth from
2015 to 2050 (See Illustration case 3).

*The increase in normalised drinking water supply volume as supplied by Vitens between 2015 and 2019 is 4.5% (Fig. 3). Due to this recent demand growth the reserve capacity within the existing drinking water supply infrastructure is already limited. The drinking water demand growth rate of the period 2015-2019 compares to the growth rate in the maximum delta scenario of 35% growth from 2015 to 2050 (Fig. 3). If this growth rate is not tempered through a significant reduction of the drinking water use, this would require a large extension of the drinking water supply infrastructure.*

***Figure 3*** *Development of the normalised annual drinking water volume supplied by Vitens (drinking water supplier), the Netherlands 2003-2019 (Van Engelenburg et al., 2020b), compared to the projected Delta scenarios on drinking water demand growth (Wolters et al., 2018), ranging between a decrease of 10% to an increase of 35% in 2050 compared to 2015. The normalized annual drinking water supply volume excludes the impact of extreme weather conditions on the actual supplied annual volumes of drinking water.*

If this strong growth rate continues, this will put serious pressure on the drinking water supply. This will partially be due to limitations in the technical infrastructure, but also partially due to limitations in the water resource availability, caused by insufficient abstraction permits, or a possibly negative impact to the hydrological system and stakeholders. Given the inflexibility of drinking water supply infrastructure to change, an integrated strategy is necessary to meet this uncertain development of the drinking water demand. To find sustainable solutions for

the future not only the technical infrastructure aspects must be solved. It also requires
strategies on water saving, expansion of permits, development of new abstraction concepts
using other water resources, as well as stakeholder processes in the design and use of the
local drinking water supply system. This case is basically an extension to the first two cases:
the growing water demand amplifies the aspects caused by the drought in 2018 and the
groundwater quality development. The results of the analysis of this case study are presented
in App. C.

# 3    Sustainability characteristics of drinking water supply

In this section the sustainability characteristics are presented, each elaborated further into
five sustainability criteria. A detailed description of the resulting sustainability criteria can be
found in Appendix E.

## 3.1    Hydrological sustainability characteristics

Three hydrological sustainability characteristics are proposed that summarise the hydrological
aspects affecting the drinking water supply as found in the case studies: *water quality*, *water*
*resource availability* and *impact of drinking water abstraction* (Table 1).
*Water quality* includes the monitoring and evaluation of current water quality, and the trends
and expected future development of the water quality and emerging contaminants, as
described in the case "Groundwater quality development". In the WHO Guidelines for
Drinking-Water Quality (WHO, 2017a) additionally the importance of microbial aspects as a
global water quality aspect with a health impact is monitored, such as bacteriological
contamination due to untreated waste water or emergencies. The WHO Guidelines for
Drinking-Water Quality (WHO, 2017a) also requires monitoring of water quality aspects
without health impact, such as salinization, water hardness, and colour, which affect the
acceptability of the drinking water (WHO and UNICEF, 2017).
*Table 1* *Summary of proposed hydrological sustainability characteristics, hydrological aspects from*
*case studies (see App. A-C),  relevant SDG[1] indicators and WHO Guidelines for Drinking-Water Quality*
*(WHO, 2017a) aspects, and hydrological sustainability criteria.*

| Hydrological sustainability characteristics | Water quality | Water resource availability | Impact of drinking water abstraction |
|---|---|---|---|
| Sustainability aspects from case studies | Monitoring and evaluation<br>Sources of pollution<br>Contaminants<br>Emerging contaminants<br>Groundwater quality<br>Surface water quality<br>Raw water quality | Other water resources<br>Surface water quantity<br>Groundwater quantity<br>Vulnerability of the water system<br>Drought impact<br>Water discharge | Impact of abstraction Groundwater levels<br>Abstraction volume<br>Balance between annual recharge and annual abstraction<br>Hydrological compensation |
| SDG 6 targets[1] | 6.3, 6.5 | 6.4, 6.5 | 6.4, 6.6 |
| WHO Guidelines for Drinking-Water Quality (WHO, 2017a) | Health risks from microbial contamination<br>Acceptability of the drinking water (salinization, hardness, colour) | Small- or large-scale emergencies caused by natural hazards, such as droughts, floods, earthquakes or forest fire | - |
| Sustainability criteria | Current raw water quality<br>Chemical aspects of water quality<br>Microbial aspects of water quality<br>Acceptability aspects of water quality<br>Monitoring and evaluation of water quality trends | Surface water quantity<br>Groundwater quantity<br>Other available water resources<br>Vulnerability used water system  for contamination<br>Natural hazards and emergencies risk | Impact on surface water system<br>Impact on groundwater system<br>Balance between annual recharge and abstraction<br>Hydrological compensation<br>Spatial impact of abstraction facility/ storage/reservoir |

[1] SDG = Sustainable Development Goal; see App. V for summary of Sustainable Development Goal 6 targets and
indicators related to sustainability characteristics (UN, 2015)
*Water resource availability* for drinking water supply can be differentiated into the surface
water and groundwater availability, as illustrated in Case 1 "2018 Summer drought". Other
sustainability aspects are the vulnerability of the surface and/or groundwater system to the
water quality being affected permanently by land use, as illustrated in the case "Groundwater
quality development". The water resource availability can also be limited due to small- or
large-scale emergencies caused by natural hazards, such as droughts, floods, earthquakes or
forest fires (WHO and UNICEF, 2017), that will put the sustainability of the local drinking water
supply under pressure.
The *impact of the drinking water abstraction* to the hydrological system entails the impact to
both the surface water system and the groundwater system, but also the balance between
the annual drinking water abstraction volume and the annual recharge of the (local) water
system. Whether the impact of the abstraction is or can possibly by compensated
hydrologically is another sustainability aspect. The spatial impact of the local drinking water
abstraction facility may also be a sustainability aspect: a drinking water facility requires a
certain water storage area or reservoir, which might have a significant spatial impact in the
area and thus might affect local stakeholders.

## 3.2   Technical sustainability characteristics

Three technical sustainability characteristics are proposed that summarise the technical
aspects for the drinking water supply as found in the case studies: *reliability* and *resilience of*
*the technical infrastructure* and *energy use and environmental impact* of the drinking water
supply (Table 2).
The *reliability* of the supply system is defined in this research as "the (un)likeliness of the
technical system to fail" (Hashimoto et al., 1982). The current technical state of the drinking
water production facility and the distribution infrastructure, and the complexity of the water
treatment are important technical sustainability criteria for the local drinking water supply
system. Other technical criteria that should be considered are the supply continuity of the
facility, which stands for the capability to meet the set legal standards for drinking water
supply under all circumstances, and the operational reliability, to solve technical failures
without disturbance of the drinking water supply.
***Table 2*** *Summary of proposed technical sustainability characteristics, technical aspects from case*
*studies (see App. A-C), relevant SDG[1] indicators and WHO Guidelines for Drinking-Water Quality*
*(WHO, 2017a) aspects, and technical sustainability criteria.*

| Technical sustainability characteristics | Reliability of technical infrastructure | Resilience of technical infrastructure | Energy use and environmental impact |
|---|---|---|---|
| **Sustainability aspects from case studies** | Drinking water pressure<br><br>Drinking water treatment<br><br>Reliability of abstraction, treatment and distribution infrastructure | Abstraction capacity<br><br>Treatment capacity<br><br>Treatment methods<br><br>Distribution capacity<br><br>Resilience of technical infrastructure | Energy use<br><br>Environmental impact<br><br>Additional excipients<br><br>Wastewater<br><br>Waste materials |
| **SDG 6 targets[1]** | 6.1, 6.4 | 6.1, 6.4 | 6.4 |
| **WHO Guidelines for Drinking-Water Quality (WHO, 2017a)** | Safely managed drinking water services, i.e. improved drinking water source on premises, available when needed and free from contamination | Resilient technologies and processes<br><br>Upgrades of water treatment and storage capacity | Reliability of the energy supply<br><br>Renewability of the energy |
| **Sustainability criteria** | Technical state abstraction and treatment facility<br><br>Technical state distribution infrastructure<br><br>Complexity of water treatment<br><br>Supply continuity for customers<br><br>Operational reliability | Abstraction permit compared to annual drinking water demand<br><br>Production capacity compared to peak demand<br><br>Flexibility of treatment method<br><br>Technical innovations to improve resilience<br><br>Technical investments to improve resilience | Energy use of abstraction and treatment<br><br>Energy use of distribution<br><br>Environmental impact (additional excipients, wastewater, waste materials)<br><br>Reliability energy supply<br><br>Use of renewable energy |

[1] SDG = Sustainable Development Goal; see App. V for summary of Sustainable Development Goal 6 targets and
indicators related to sustainability characteristics (UN, 2015)
In this research the *resilience* of the drinking water supply system is defined as "the possibility
to respond to short- and long-term changes in water demand or water quality" (Hashimoto et
al., 1982). Climate change and other developments in water demand and quality call for the
use of more resilient technologies and processes, and may require upgrades of water
treatment and storage capacity (WHO and UNICEF, 2017). The cases "2018 Summer drought"
as well as "Drinking water demand growth" emphasise the importance of the available
abstraction permits, and treatment and distribution capacity compared to the annual and
peak water demand respectively for the resilience of the local drinking water supply system.
Furthermore, the flexibility of the treatment method determines whether a drinking water
supply system can deal with variation in, or deterioration of water quality and emerging
contaminants, the sustainability aspects found in the case "Groundwater quality
development".
*Energy use and environmental impact* includes the sustainability aspects from the cases
"Groundwater quality development" and "Drinking water demand growth": the energy use of
abstraction, treatment and distribution, and the environmental impact of additional
excipients, waste water and other waste products of the treatment. Especially when the raw
water quality deteriorates, the required water treatment methods become more complex. In
general, this leads to large investments, as well as an increasing energy use and environmental
impact, e.g. when advanced membrane filtration methods are required. Additional global
sustainability aspects are the reliability of the energy supply, and the renewability of the
energy that is used (WHO, 2017a).

### 3.3 Socio-economic sustainability characteristics

Three socio-economic sustainability characteristics are proposed that summarise the socio-
economic aspects affecting the drinking water supply as found in the case studies: *drinking
water availability, water governance*, and *land and water use* (Table 3).
The *drinking water availability* can be quantified by the percentage of households connected
to the drinking water supply. A sustainable local drinking water supply provides sufficient
drinking water of a quality that meets the national or international drinking water standards,
for a tariff that is affordable to all households (UN, 2015). In the Netherlands the drinking
water tariff by law must be built on a cost-recovery, transparent and non-discriminatory basis
(Dutch Government, 2009). Water saving strategies will reduce the drinking water demand
growth and therefore will contribute to the sustainability. Drinking water safety is a
prerequisite for public health and sustainable drinking water supply. The WHO Guidelines
consider water safety plans essential to provide the basis for system protection and process
control to ensure water quality issues present a negligible risk to public health and that the
drinking water is acceptable to consumers. Therefore WHO Guidelines for Drinking-Water
Quality (2017) monitors the availability of water safety plans including emergency plans on
how to act in case of drinking water supply disturbances, shortages, or drinking water quality
emergencies (WHO and UNICEF, 2017). A water safety plan can be built on various safety
protocols.
*Water governance* focuses on policies and legislation, enforcement and compliance of
regulations. Good governance also includes decision-making processes considering different
stakeholder interests, to ensure accountable, transparent and participatory governance
(UNESCAP, 2009). The availability of (inter)national and local policies and legislation on
drinking water supply as well as on water management, including regulations and permits,
and the level of compliance of the drinking water supplier to these policies and legislation, are
important for the socio-economic sustainability. The sustainability of local drinking water
supply is also characterised by the stakeholders' interests related to the presence of a local
drinking water abstraction, and by how local authorities weigh these interests in their
decision-making processes. A final aspect in water governance that reaches further than local
stakeholder interests is the risk of small- or large-scale emergencies for the drinking water
supply caused by human activities or conflicts (WHO and UNICEF, 2017).
The local *land and water use*, at surface and subsurface level, affects the water quality and
quantity. It may have resulted in historical contaminant sources, causing point or non-point
water pollution, but it may also lead to emerging contaminants that provide new risks to water
quality. Additionally, water use for other purposes may limit the availability of water
resources for drinking water. Regulations to protect water quality or water quantity may cause
limitations for local land and water use. Financial compensation for suffered economic
damage due to the impact of the abstraction or the limitations caused by protection
regulations can be an important aspect for the sustainability of the drinking water supply
system.
*Table 3* *Summary of proposed socio-economic sustainability characteristics, socio-economic aspects*
*from case studies (see App. A-C), relevant SDG[1] indicators and  WHO Guidelines for Drinking-Water*
*Quality (WHO, 2017a) aspects, and socio-economic sustainability criteria.*

| Socio-economic sustainability characteristics | Drinking water availability | Water governance | Land and water use |
|---|---|---|---|
| **Sustainability aspects from case studies** | Customers<br>Drinking water availability<br>Drinking water demand<br>Drinking water tariff<br>Drinking water quality<br>Drinking water volume<br>Drinking water shortage<br>Emergencies, disturbances<br>Water saving | Abstraction permits<br>Drinking water standards<br>Water authorities<br>Water legislation, policy and regulations<br>Drinking water suppliers<br>Compliance<br>Stakeholders | Water use<br>Land use<br>Agriculture<br>Nature, groundwater-dependent ecosystems<br>Financial compensation<br>Spatial impact |
| **SDG 6 targets[1]** | 6.1 | 6.3, 6.4, 6.5, 6.6, 6.a, 6.b | 6.3, 6.4 |
| **WHO Guidelines for Drinking-Water Quality (WHO, 2017a)** | Water safety plan | Small- or large-scale emergencies for the drinking water supply caused by human activities or conflicts | - |
| **Sustainability criteria** | Percentage connected households<br>Drinking water service quality<br>Drinking water tariff<br>Water saving strategy<br>Water safety protocols | Availability of (drinking) water legislation and policies<br>Compliance of drinking water supplier<br>Decision-making process by (local) authorities<br>Local stakeholder interests<br>Emergency risk caused by human activities or conflicts | Land use (including subsurface use)<br>Water use for other purposes than drinking water<br>Regulations on land and water use<br>Limitations to land or water use<br>Financial compensation of economic damage from impact of abstraction or limitations to land use |

[1] SDG = Sustainable Development Goal; see App. V for summary of Sustainable Development Goal 6 targets and
indicators related to sustainability characteristics (UN, 2015)

# 4 Discussion

## 4.1 Use of DPSIR systems approach

In this study we used an integrated systems approach to analyse the local drinking water supply system, combining hydrological, technical and socio-economic aspects of the system. The analysis of the three selected cases with DPSIR supported the identification of aspects that shape the sustainability of the local drinking water supply system. The case analysis did indeed help to account for differences between short-term and long-term developments, and for the impact of external influences that come from the national and international scale.

The applied DPSIR approach is a linear socio-ecological framework originally developed to identify the impact of human activities on the *state* of the environmental system (Binder et al., 2013). However, the local drinking water supply system is a complex system rather than linear, because the *impact* of a *pressure* to one system element could present a *pressure* to another system element. This complicated the identification of *pressures* and *impacts*. For instance, high temperatures and lack of precipitation caused a higher drinking water demand and surface water quality deterioration. Both consequently presented *pressures* with an *impact* to the resilience and reliability of the technical drinking water supply infrastructure. Although this hampered the analysis, the use of DPSIR supported a systematic analysis of the local drinking water supply cases and helped to identify the sustainability aspects. Use of a different integrated systems approach would not have led to a significantly different outcome of the case analysis. A next step could potentially be to use the identified system characteristics for a system dynamics analysis and modelling. However, this is beyond the scope of this current research.

## 4.2 General applicability of the sustainability characteristics

To increase the general applicability of the results from the analysis of the Dutch cases on drinking water supply, the identified sustainability aspects were related to worldwide acknowledged sustainability aspects, by cross-checking with international policies on drinking water supply. This put the aspects in a broader perspective, which may contribute to the transferability of the proposed sustainability characteristics and criteria to other areas.

Assessments to understand the sustainability challenges as well as the impact of future developments and adaptation options are seen as powerful tools for policy making (Ness et al., 2007, Singh et al., 2012). The sustainability characteristics as proposed in this research may be used to develop a sustainability assessment for the local drinking water supply system, that can help to identify sustainability challenges and trade-offs of adaptation strategies. Trade-off analysis supports decision-making processes and makes these processes more transparent to local stakeholders (Hellegers and Leflaive, 2015). Based on the local situation and data availability, adequate indicators and indices can be selected to quantify the sustainability characteristics in a certain area (Van Engelenburg et al., 2019).

## 5 Conclusions

The aim of this study was to identify a set of characteristics that describe the sustainability of a local drinking water supply system in the Netherlands to support policy- and decision-making on sustainable drinking water supply. The use of the DPSIR systems approach was an adequate method for the analysis of the cases. The results of the analysis of the three cases confirmed the hypothesis that sustainability is contextual, resulting in different sustainability aspects in the various cases. The combined results of the analysis of three different practice cases contributed to a better understanding of drinking water supply in the Netherlands. Cross-

checking of the results of case analysis with international policies on drinking water supply
provided a wider context than the Netherlands and has thus contributed to the general
applicability of the identified sustainability characteristics.
Based on the presented analysis, the following set of hydrological, technical and socio-
economic sustainability characteristics is proposed, respectively: (1) *water quality*, *water*
*resource availability*, and *impact of drinking water abstraction*; (2) *reliability* and *resilience of*
*the technical system*, and *energy use and environmental impact*; (3) *drinking water availability,*
*water governance*, and *land and water use*. Elaboration of the sustainability characteristics
into more detailed criteria may further increase the value of the results of this research in the
process of development of policies on sustainable drinking water supply in the Netherlands.
**Data availability:** The source data used for the illustrations of the cases are available at
request.
**Author Contributions:** Conceptualization J.v.E., P.J.G.J.H., E.v.S.; methodology J.v.E.,
P.J.G.J.H., E.v.S.; data curation J.v.E.; investigation J.v.E.; writing – original draft preparation
J.v.E.; writing – review and editing J.v.E., P.J.G.J.H., E.v.S., A.J.T., R.U.; visualization J.v.E.;
supervision P.J.G.J.H., E.v.S., A.J.T., R.U. (See CRediT taxonomy for term explanation).
**Acknowledgments:** We are indebted to Vitens staff Maarten Fleuren and Martin de Jonge for
their data collection for the illustrations of the analysed drinking water supply cases, and to
Vitens staff Mark de Vries, Henk Hunneman and Rian Kloosterman for cross-checking the
results of the case analysis. We thank the two anonymous referees for their comments that
helped to further improve the manuscript.

Conflicts of Interest: The first author performed the research partially in time funded by

Vitens, where she is employed. She had carte blanche for the content of the research.

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

# Appendix A    Results of analysis case 1 "2018 Summer drought"

*Table A.1* Summary of impact, short-term and long-term response and sustainability aspects in case 1 "2018 Summer drought" (for complete results of the case study see Table A.2).

| Impact | Short-term response | Long-term response | Sustainability aspects |
|---|---|---|---|
| Extreme drinking water use, high drinking water demand. | Drinking water suppliers increased abstraction volume. | Development of water saving strategies. | Drinking water use, drinking water demand, drinking water suppliers, abstraction volumes, water saving. |
| Drought, falling water discharges and groundwater levels, damage to groundwater-dependent ecosystems and agriculture. | Water use limitations, water authorities applied existing drought water policy, risk for water quality. | Development of additional water shortage policy for water management and water governance. | Drought, water discharge, groundwater levels, groundwater-dependent ecosystems, agriculture, water use, water authorities, water policy, water management, water governance, water availability. |
| Customers worried about drinking water availability. | Drinking water suppliers called upon customers for drinking water saving. | Societal support for drinking water saving strategies. | Customers, drinking water availability, drinking water suppliers, water saving. |
| Declining surface water discharge and quality. | Drinking water supplies took measures to safeguard raw water quality. | Development of additional policies on water quality protection. | Surface water discharge, surface water quality, drinking water suppliers, raw water quality, water management policies, water use. |
| Groundwater quality deterioration. | No response possible due to lack of water. | Development of additional policies on water quality protection. | Groundwater quality, surface water quality, water shortage, surface water discharge, water management policies |
| Drinking water quality at risk due to rising water temperature in pipelines. | Sufficient refreshment due to high demand. | Changing the design standard of distribution pipelines to limit risk of temperature rise. | Drinking water quality, treatment method, distribution infrastructure. |
| Increasing abstraction volume, resulting in increasing impact to land use. | Stakeholder complaints by agriculture and nature. | Increased societal pressure on reduction of impact of drinking water abstraction. | Drinking water demand, abstraction volume, impact of abstraction, land use, stakeholders, agriculture, nature, drinking water suppliers. |
| Exceedance of abstraction permits, limiting the resilience of the technical infrastructure. | Enforcement procedures by legal authorities. | Extension of drinking water abstraction permits and water saving strategies. | Drinking water demand, abstraction volume, abstraction capacity, abstraction permit, resilience of abstraction, legal authorities, water regulations, water legislation, drinking water saving. |
| Shortage of drinking water during peak demand due to insufficient resilience of treatment infrastructure. | Reduced drinking water supply volume. | Adjustment of resilience and reliability of treatment infrastructure. | Treatment volume, treatment capacity, drinking water shortage, reliability of the treatment, resilience of the treatment, drinking water standards, drinking water demand, drinking water suppliers. |
| Insufficient distribution capacity | Lowering drinking water pressure to reduce drinking water volume | Adjustment of resilience and reliability of distribution infrastructure. | Distribution capacity, resilience and reliability of distribution, drinking water suppliers, drinking water volume, drinking water standards. |
| Major disturbances could cause a serious disruption of the supply. | Maximum personnel deployment by drinking water suppliers. | Investments to improve resilience and reliability of technical infrastructure by drinking water suppliers. | Drinking water demand, reliability of technical infrastructure, drinking water suppliers. |

| Impact | Short-term response | Long-term response | Sustainability aspects |
|---|---|---|---|
| High energy use and environmental impact of extreme drinking water production. | - | Incorporating impact to energy use and environmental impact in design of measures to improve resilience and reliability of technical infrastructure. | Drinking water demand, energy use, environmental impact, drinking water suppliers. |

*Table A.2* Results analysis of Case 1 "2018 summer drought". For each pressure the response and impacts to the state of the local drinking water supply system are described. The sustainability aspects in the case are displayed in bold. The grey cells refer to Table A.1.

| Driver | Pressure | Impact | Short-term response | Long-term response | Sustainability aspects |
|---|---|---|---|---|---|
| Extreme weather event | High temperature, high evaporation, no precipitation | Extreme drinking water use, high drinking water demand. | Drinking water suppliers increased abstraction volume. | Development of water saving strategies. | Drinking water use, drinking water demand, drinking water suppliers, abstraction volume, water saving. |
| | | The summer affected the **drinking water use**: filling swimming pools, watering gardens, extra showering all together led to a very high **drinking water demand**. Additionally, there also were requests from concerned citizens for applying drinking water to refill ponds that fell dry due to the extreme drought. | **Drinking water suppliers** increased the **abstraction volume** to meet the increased drinking water demand. | The drought (re-)initiated a discourse on **water saving strategies**, including controversial measures such as progressive drinking water tariffs and differentiation in high-grade (household and sanitation, food production) and low-grade (pools, gardens, process water) use. | |
| Extreme weather event | High evaporation, no precipitation | Drought, falling water discharges and groundwater levels, damage to groundwater-dependent ecosystems and agriculture. | Water use limitations, water authorities applied existing drought water policy, risk for water quality. | Development of additional water shortage policy for water management and water governance. | Drought, water discharge, groundwater levels, groundwater-dependent ecosystems, agriculture, water use, water authorities, water policy, water management, water governance, water availability. |
| | | The **drought** caused falling **water discharges** and **groundwater levels**: river discharges declined, springs and brooks fell dry, and vegetation withered or even died due to low groundwater levels and high temperatures. **Groundwater-dependent ecosystems** such as wetlands as well as **agriculture** produce suffered from the drought. | Limitations to **water use** from water system. **Water authorities** applied the special **water policy** that was developed for periods with low water availability. Drinking water supply has a high ranking because of its high societal relevance. In some ecologically vulnerable areas, there is a **water policy** to resolve local surface water shortages by supplementing from larger water bodies such as rivers. This affects the local **surface water quality** and may also affect the **groundwater quality**. | Discourse and policy development on **water management** and **water governance** aiming at a further prioritisation and limitations of water use during water shortage, and retention of surface water and groundwater during periods with sufficient **water availability.** | |

| Driver | Pressure | Impact | Short-term response | Long-term response | Sustainability aspects |
|---|---|---|---|---|---|
| Extreme weather event | High evaporation, no precipitation | Customers worried about drinking water availability. | Drinking water suppliers called upon customers for drinking water saving. | Societal support for drinking water saving strategies. | Customers, drinking water availability, drinking water suppliers, water saving. |
| | | Because of the visible damage to vegetation due to the drought, **customers** started to worry about the **drinking water availability.** | **Drinking water suppliers** communicated that there still was sufficient drinking water, but people were asked to spread the drinking water use to reduce the peak demand. Later that summer **customers** were called for **water saving.** | The drought raised awareness under customers that there are limits to the **drinking water availability**, thus creating (some) societal support for (drinking) water saving. | |
| Extreme weather event | No precipitation | Declining surface water discharge and quality. | Drinking water supplies took measures to safeguard raw water quality. | Development of additional policies on water quality protection. | Surface water discharge, surface water quality, drinking water suppliers, raw water quality, water management policies, water use. |
| | | Due to the lack of rain, the share of industrial wastewater and treated sewage water to the **surface water discharge** increased, which caused the **water quality** in surface waters deteriorated. | **Drinking water suppliers** that use surface water as resource took measures to safeguard the **raw water quality.** | The **surface water discharge and quality** problems may induce development of **water management policies** that aim to reduce the impact of treated sewage and industrial wastewater, by reduction of **water use** or improvement of treatment. | |
| Extreme weather event | Declining surface water quality | Groundwater quality deterioration. | No response possible due to lack of water. | Development of additional policies on water quality protection. | Groundwater quality, surface water quality, water shortage, surface water discharge, water management policies. |
| | | The impact of an incidental warm and dry summer to the groundwater quality is limited, but when comparable droughts will happen frequently the **groundwater quality** may deteriorate due to the impact of a declining **surface water quality.** | In some surface water bodies refreshment was required to guard the **surface water quality**, but due to the lack of precipitation there was **a water shortage**, so insufficient water was available for this refreshment. | The fact that **surface water discharge** and **quality** may affect **groundwater quality** supports the need of **water management policies** that aim to refresh water bodies and to reduce the impact of treated sewage and industrial wastewater. | |
| Extreme weather event | High temperature | Drinking water quality at risk due to rising water temperature in pipelines. | Sufficient refreshment due to high demand. | Changing the design standard of distribution pipelines to limit risk of temperature rise. | Drinking water quality, treatment method, distribution infrastructure. |
| | | The extreme temperatures led to an increased surface water temperature, and soil temperature, that may have affected drinking water temperature in distribution infrastructure. This introduces a **drinking water quality** risk. | When surface water is the main resource for drinking water, the **water quality** risk will be limited by a **treatment** method that ensures the bacteriological quality of the drinking water. Sufficient refreshment within storage and high stream velocities in pipelines reduce the risk of temperature rise in the **distribution infrastructure.** | The risk of **drinking water quality** aspects caused by increased drinking water temperature due to climate change may have consequences for the design of the **distribution infrastructure**. | |
| | High drinking water demand | Increasing abstraction volume, resulting in increasing impact on land use. | Stakeholder complaints by agriculture and nature. | Increased societal pressure on reduction of impact of drinking water abstraction. | Drinking water demand, abstraction volume, |

| Driver | Pressure | Impact | Short-term response | Long-term response | Sustainability aspects |
|--------|----------|--------|---------------------|--------------------|------------------------|
| Extreme weather event | | To meet the high **drinking water demand**, the **abstraction volume** rose to a high level. In some local areas the **impact of the abstraction** added up with the extreme drought and high temperatures, affecting the **land use.** | **Stakeholders** on agriculture and nature complained about the impact of the extra abstraction to their land use. | The drought impact enlarged the societal pressure to **drinking water suppliers** to reduce the **impact** of local **drinking water abstraction** to the water system. | impact of abstraction, land use, stakeholders, agriculture, nature, drinking water suppliers. |
| Extreme weather event | High drinking water demand | Exceedance of abstraction permits, limiting the resilience of the technical infrastructure. | Enforcement procedures by legal authorities. | Extension of drinking water abstraction permits and water saving strategies. | Drinking water demand, abstraction volume, abstraction capacity, abstraction permit, resilience of abstraction, legal authorities, water regulations, water legislation, drinking water saving. |
| | | To meet the high **drinking water demand**, the **abstraction volume** rose to a high level. The available **abstraction capacity** combined with the high **abstraction volumes** led to exceedance of the **abstraction permits**. Some local drinking abstractions exceeded the monthly permitted volume, and some abstractions even exceeded the yearly permitted volume, failing drinking water regulations. This compromised the **resilience of the abstractions.** | **Legal authorities** (provinces and water boards) started enforcement procedures to meet the **water regulations**. The legal authority urged the drinking water supplier to stay within these limits. However, the drinking **water legislation** also had to be met to ensure continuous supply of good quality drinking water at all times. | The exceedance of **abstraction permit limits** set off enforcement actions by the government, resulting in an increased need for additional **abstraction permits**, as well as **drinking water saving** strategies to reduce the **drinking water demand.** | |
| Extreme weather event | High peak demand for drinking water | Shortage of drinking water during peak demand due to Insufficient resilience of treatment infrastructure. | Reduced drinking water supply volume. | Adjustment of resilience and reliability of treatment infrastructure. | Treatment volume, treatment capacity, drinking water shortage, reliability of the treatment, resilience of the treatment, drinking water standards, drinking water demand, drinking water suppliers. |
| | | To meet the high peak demand, the **treatment volume** rose to a high level. In some parts of the drinking water supply there was insufficient **treatment capacity**, causing a temporary **shortage of drinking water** during peak demand, compromising **the reliability of the treatment**. These limitations showed that the treatment is not **resilient** for this extreme peak demand. | There is no response available when the treatment capacity is insufficient, except reducing the drinking water supply volume. Exceeding the treatment capacity (by e.g. increasing the filter flow velocity or reducing the cleansing frequency of the filters) would introduce the risk of not meeting the **drinking water standards.** | The drought identified various locations in the technical infrastructure where the **treatment capacity** was not reliable at **peak drinking water demand**, which set **drinking water suppliers** off to solve these local treatment aspects. To adjust all aspects will take several years. | |
| Extreme weather event | High peak demand for drinking water | Insufficient distribution capacity. | Lowering drinking water pressure to reduce drinking water volume. | Adjustment of resilience and reliability of distribution infrastructure. | Distribution capacity, resilience and reliability of distribution, drinking water suppliers, drinking |
| | | In some parts of the drinking water supply there was insufficient **distribution** | To reduce the drinking water volume that was supplied, **drinking water suppliers** | The drought identified locations in the technical infrastructure where the | water |

| Driver | Pressure | Impact | Short-term response | Long-term response | Sustainability aspects |
|---|---|---|---|---|---|
| | | **capacity** due to hydraulic limitations, insufficient storage capacity, or age and quality of the pipelines. In some areas this caused unintended low drinking water pressures. These limitations put the **reliability** of the **distribution** under pressure and showed that the distribution capacity was not **resilient** for this extreme peak demand. | lowered the drinking water pressure intendedly in some areas. The impact of this pressure reduction is a decreased **drinking water volume** from taps. By reducing drinking water pressure, the distributed drinking water volume was reduced, however this also led to falling short of the mandatory **drinking water standards** in some areas. | **distribution capacity** was not reliable at **peak demand**, which set **drinking water suppliers** off to solve these local distribution aspects. To adjust all aspects will take several years. | water volume, drinking water standards. |
| Extreme weather event | High peak demand for drinking water | Major disturbances could cause a serious disruption of the supply. | Maximal personnel deployment by drinking water suppliers. | Investments to improve resilience and reliability of technical infrastructure by drinking water suppliers. | Drinking water demand, reliability of technical infrastructure, drinking water suppliers. |
| | | The high **peak demand** required a maximal exploitation of the **technical infrastructure**. To ensure the **reliability of the drinking water supply**, many parts of the infrastructure are designed redundant, which limits the impact of disturbances for customers. However, a major disturbance in the infrastructure, such as failure of a large transportation pipeline, could have led to disruption of the supply, because the resilience was limited due to limited reserve capacity and reduced maintenance during the extreme drinking water demand period. | To ensure the **reliability of the drinking water supply**, disturbances are always solved with priority. During the extreme peak period **drinking water suppliers** had all personnel put on standby to immediately solve any disturbances. | The drought identified locations in the technical infrastructure where not reliable at **peak demand**, which set **drinking water suppliers** off to solve these local aspects, and where necessary create redundancy to decrease the risk of disturbances, and thus improve the **reliability.** | |
| Extreme weather event | High peak demand for drinking water | High energy use and environmental impact of extreme drinking water production. | - | Incorporating impact on energy use and environmental impact in design of measures to improve resilience and reliability of technical infrastructure. | Drinking water demand, energy use, environmental impact, drinking water suppliers. |
| | | The magnitude and duration of the **peak demand** forced a maximal exploitation of the technical infrastructure, causing a maximal **energy use** and **environmental impact.** | There was no short-term response available to reduce the energy use and environmental impact. | The drought identified locations in the technical infrastructure where not reliable at **peak demand**, which set **drinking water suppliers** off to solve these local aspects. **Energy use** and **environmental impact** are important aspects that are considered in the design of the solutions for these aspects. | |

# Appendix B    Results of analysis case 2 "Groundwater quality development"

**Table B.1** *Summary of impact, short-term and long-term response and sustainability aspects in case 2, "Groundwater quality development" (for complete results of the case study see Table B.2).*

| Impact | Short-term response | Long-term response | Sustainability aspects |
|---|---|---|---|
| Surface water quality deteriorates due to limited surface water discharge. | Monitoring and evaluation of water quality development. | Water legislation on water quality and quantity protection, drinking water savings strategies. | Surface water quality, surface water discharge, monitoring and evaluation, water legislation, water quality and quantity, drinking water saving. |
| Groundwater quality deteriorates due to deteriorating surface water quality. | Monitoring and evaluation of water quality development. | Improvement of sewage and waste-water treatment, and water saving strategies. | Groundwater quality, surface water quality, monitoring and evaluation, water saving. |
| Soil energy systems may affect groundwater quality. | Monitoring and evaluation of water quality development, research. | Groundwater protection regulations. | Groundwater quality, groundwater pollution, research, monitoring and evaluation, regulations, groundwater quality protection. |
| Local and upstream land and water use affects the surface water quality. | Monitoring and evaluation of water quality development. | Policy and measures to meet water legislation to protect and improve water quality and quantity. | Surface water quality, land and water use, contaminants, monitoring and evaluation, water legislation, water quantity. |
| Diffuse and point sources of pollution affect surface water and groundwater quality. | Monitoring and evaluation of water quality development. | Measures to remove historical sources of pollution and to prevent new sources of pollution. | Groundwater quality, nutrients, organic micro-pollutants, other contaminants, surface water quality, monitoring and evaluation, water legislation, water quality protection. |
| Emerging contaminants in surface and groundwater require new drinking water treatment methods. | Enforcement of groundwater protection regulations on pollution incidents and monitoring and evaluation. | Development of treatment methods to remove emerging contaminants from sewage, industrial wastewater and/or drinking water. | Emerging contaminants, groundwater quality, surface water quality, resilience and reliability of the drinking water treatment, groundwater protection, land and water use, water legislation, sources of pollution, drinking water treatment methods, energy use, environmental impact, drinking water tariff. |
| Land use (change) may cause groundwater quality deterioration. | Enforcement of groundwater protection regulations on land use change and monitoring and evaluation. | Combination of extensive land use functions with drinking water abstraction. | Land use change, groundwater quality, sources of pollution, groundwater protection regulations, water use, enforcement of regulations, monitoring and evaluation, drinking water abstraction, extensive land use, nature, agriculture, water system. |
| Surface water and groundwater quality deterioration determine the required drinking water treatment. | Monitoring of drinking water quality, in case of emergencies measures are taken to safeguard the drinking water quality. | Adjustment of treatment methods to be able to continue to meet the drinking water standards. | Raw water quality, drinking water standards, water quality, vulnerability of the water system for contamination, treatment methods, reliability and resilience of treatment, drinking water quality, emergencies, energy use, environmental impact, drinking water tariff. |
| Variations in raw water quality can only be handled if treatment method is resilient to these variations. | Monitoring and evaluation of water quality development. | Increase of resilience and reliability of drinking water treatment. | Surface water quality, groundwater quality, resilience and reliability of the treatment, monitoring and evaluation, raw water quality, energy use, environmental impact, drinking water tariff. |

**Table B.2** *Results analysis of Case 2 "Groundwater quality development". The sustainability aspects in this case are displayed in bold. The grey cells refer to Table B.1.*

| Drivers | Pressure | Impact | Short-term response | Long-term response | Sustainability aspects |
|---|---|---|---|---|---|
| Changing climate variability | Less summer precipitation, higher summer temperatures | Surface water quality deteriorates due to limited surface water discharge. | Monitoring and evaluation of water quality development. | Water legislation on water quality and quantity protection and drinking water savings strategies. | Surface water quality, surface water discharge, monitoring and evaluation, water legislation, water quality and quantity, drinking water saving. |
| | | In summer **surface water quality** deteriorates due to limited **surface water discharge**, combined with increasing contribution of industrial and treated sewage water recharges compared to natural discharges due to lack of summer precipitation. | **Monitoring and evaluation** of water quality development is necessary to be able to timely respond to a changing surface water quality. | Land and water use must meet **water legislation** as set by the European Water Framework Directive and national water legislation to protect and improve **water quality and quantity**. Further improvement of sewage and wastewater treatment will reduce the impact on the **surface water quality**. **Drinking water saving** strategies can also lead to reduction of treated sewage water recharges and industrial recharges. | |
| Changing climate variability | Surface water quality deterioration | Groundwater quality deteriorates due to deteriorating surface water quality. | Monitoring and evaluation of water quality development. | Improvement of sewage and wastewater treatment, and water saving strategies. | Groundwater quality, surface water quality, monitoring and evaluation, water saving. |
| | | **Groundwater quality** may be affected by the deteriorating **surface water quality** during summer periods through natural or artificial infiltration of surface water. | **Monitoring and evaluation** of water quality development is necessary to be able to timely respond to a changing **surface water quality.** | Further improvement of sewage and wastewater treatment will reduce the impact on the **surface water quality**. (**Drinking) water saving** strategies can also lead to reduction of treated sewage water recharges and industrial recharges. | |
| Socio-economic developments | Increase in use of soil energy systems | Soil energy systems may affect groundwater quality. | Monitoring and evaluation of water quality development, research. | Groundwater protection regulations. | Groundwater quality, groundwater pollution, research, monitoring and evaluation, regulations, groundwater quality protection. |
| | | There is a transition going on towards renewable energy resources, not only wind and solar energy but also towards use of soil energy. **Groundwater quality** may be affected by the use of soil energy, due to risk of **groundwater pollution** by soil energy systems | **Research** on, and **monitoring and evaluation** of the impact of soil energy to the groundwater quality (including temperature impact) is necessary to avoid introduction of new **sources of pollution** by soil energy systems. | **Regulations** on soil energy help to limit the risk for **groundwater quality**. **Policy** is developed to exclude vulnerable groundwater systems that are used for drinking water supply for soil energy use for **groundwater quality protection**. | |

| Drivers | Pressure | Impact | Short-term response | Long-term response | Sustainability aspects |
|---------|----------|--------|---------------------|--------------------|------------------------|
| | | and the risk of leakage through aquitards that protect aquifers. | | | |
| Population growth, industrial developments | Increasing sewage and wastewater discharges | Local and upstream land and water use affects the surface water quality. | Monitoring and evaluation of water quality development. | Policy and measures to meet water legislation to protect and improve water quality and quantity. | Surface water quality, land and water use, contaminants, monitoring and evaluation, water legislation, water quantity. |
| | | **Surface water quality** is affected by local and upstream **land and water use** activities. Discharge of treated sewage water as well as industrial wastewater discharges introduce **contaminants** in the water system. | **Monitoring and evaluation** of the water quality development is necessary to be able to timely respond to a changing surface water quality. | Land and water use must meet **water legislation** as set by the European Water Framework Directive and national water legislation to protect and improve **water quality and quantity.** According to the **water legislation** in the European Water Framework Directive additional measures must be taken to reach the set goals in 2027. | |
| Population growth, industrial developments | Historical pollution, increasing sewage and wastewater discharges (change) | Diffuse and point sources of pollution affect surface water and groundwater quality. | Monitoring and evaluation of water quality development. | Measures to remove historical sources of pollution and to prevent new sources of pollution. | Groundwater quality, nutrients, organic micro-pollutants, other contaminants, surface water quality, monitoring and evaluation, water legislation, water quality protection. |
| | | **Groundwater quality** is affected by diffuse and point sources of pollution, such as **nutrients, organic micro-pollutants and other contaminants** caused by historic land and water use. Groundwater can be influenced by (historic and current) **surface water quality** through natural or artificial infiltration of surface water. | The impact of historical **contaminations** will proceed further into the groundwater system and cannot be undone, unless soil processes help to break down contaminants. **Monitoring and evaluation** are necessary to be able to timely respond to a changing **water quality.** | Historical **contaminations** from past land use will affect the **groundwater quality** for a long period of time due to the low stream velocity of groundwater. Some historical point-pollutions may be removed through soil and groundwater remediation, but diffuse pollution cannot be removed. However, according to the **water legislation** in the European Water Framework Directive additional measures must be taken to reach the set goals on **water quality protection** in 2027. | |
| Population growth, industrial developments | Increasing sewage and wastewater discharges | Emerging contaminants in surface and groundwater require new drinking water treatment methods. | Enforcement of groundwater protection regulations on pollution incidents and monitoring and evaluation. | Development of treatment methods to remove emerging contaminants from sewage, industrial wastewater and/or drinking water. | Emerging contaminants, groundwater quality, surface water quality, resilience and reliability of the drinking water treatment, groundwater protection, land and water use, water legislation, sources of pollution, drinking water |
| | | **Emerging contaminants**, such as new industrial pollutants, medicine residues and micro plastics, may pose new threats to the **groundwater and surface water quality**, and consequently the **raw** | **Groundwater protection regulations** on **land and water use** aim to reduce the risk of pollutions to avoid **groundwater quality** deterioration. This includes regulations for small incidents with point pollutions such as | According to the **water legislation** in the European Water Framework Directive known **sources of pollution** must be reduced and new **sources of pollution** must be prevented. This may include prohibition by law or measures to reduce the use of specific chemical products. | |

| Drivers | Pressure | Impact | Short-term response | Long-term response | Sustainability aspects |
|---|---|---|---|---|---|
| | | **water quality**, especially when they cannot be removed using the currently available treatment methods. The changes limit the **resilience and reliability of the drinking water treatment**. | caused by a car accident to be reported and solved immediately by removing the source of pollution. Continuous **enforcement of these regulations** is essential. **Monitoring and evaluation** are necessary to be able to timely respond to a changing **water quality.** | To deal with **emerging contaminants** it is essential to limit or remove the **contaminant source**. If all these measures fail, the contaminants must be removed by the drinking water treatment. Other or new **drinking water treatment methods** may be required. New treatment methods may cause an increase of **energy use** and **environmental impact** (excipients, wastewater, waste materials). This may lead to a higher **drinking water tariff.** | treatment methods, energy use, environmental impact, drinking water tariffs. |
| Population growth, industrial developments | Land use change | Land use (change) may cause groundwater quality deterioration. | Enforcement of groundwater protection regulations on land use change and monitoring and evaluation. | Combination of extensive land use functions with drinking water abstraction. | Land use change, groundwater quality, sources of pollution, groundwater protection regulations, water use, enforcement of regulations, monitoring and evaluation, drinking water abstraction, extensive land use, nature, agriculture, water system. |
| | | **Land use change** may cause **groundwater quality** deterioration due to the risk of diffuse of point **sources of pollution.** The impact may be limited if land use changes towards less polluting land use functions. | **Groundwater protection regulations** on **land and water use** aim to reduce the risk of pollutions to avoid **groundwater quality** deterioration. This includes regulations on land use change developments. Continuous **enforcement** of these regulations is essential. **Monitoring and evaluation** is necessary to be able to timely respond to a changing **water quality.** | Combining extensive **land use** functions such as **nature** and sustainable **agriculture** with **drinking water abstraction** In local areas to reduce the **groundwater quality** deterioration rate, depending on the land use as well as hydrological and chemical characteristics of the **water system**. | |
| Changing climate variability, population growth, industrial developments | Surface water and groundwater quality deterioration | Surface water and groundwater quality deterioration determine the required drinking water treatment. | Monitoring of drinking water quality, in case of emergencies measures are taken to safeguard the drinking water quality. | Adjustment of treatment methods to be able to continue to meet the drinking water standards. | Raw water quality, drinking water standards, water quality, vulnerability of the water system for contamination, treatment methods, reliability and resilience of treatment, drinking water quality, emergencies, energy use, environmental impact, drinking water tariffs. |
| | | The **raw water quality** of the abstracted groundwater or surface water determines the treatment that is necessary to meet the legal **drinking water standards**. When **water quality** deteriorates in general, due to the **vulnerability of the water system for contamination** different and more complex **treatment methods** | The **drinking water quality** is constantly monitored and checked with drinking water standards. In case of drinking water quality **emergencies** local measures are taken, such as temporary boiling instructions to customers or temporary additional treatment, to safeguard the drinking water quality. | A deteriorating raw water quality may require adjustment of **treatment methods** to meet the **drinking water standards** and to ensure the **resilience and reliability of the treatment**. In general, a more complex treatment method leads to a higher **energy use**, and a higher **environmental impact** due to additional use of excipients, water loss and waste materials, which will lead to a higher **drinking water tariff.** | |

| Drivers | Pressure | Impact | Short-term response | Long-term response | Sustainability aspects |
|---------|----------|--------|---------------------|--------------------|------------------------|
| | | become necessary to ensure the **reliability of the treatment** to meet the drinking water standards. The **resilience** of the treatment method or capacity may be insufficient to respond to variability in raw water quality. | | If the raw water quality is under extreme pressure, adjustment of treatment methods may not be possible. This can ultimately lead to the decision to close the local drinking water abstraction and force the drinking water supplier to find and develop a replacing abstraction location. | |
| Population growth, industrial developments | Incidental changes in surface water and groundwater quality | Variations in raw water quality can only be handled if treatment method is resilient to these variations. | Monitoring and evaluation of water quality development. | Increase of resilience and reliability of drinking water treatment. | Surface water quality, groundwater quality, resilience and reliability of the treatment, monitoring and evaluation, raw water quality, energy use, environmental impact, drinking water tariffs. |
| | | Especially **surface water quality** can show strong water quality variations. They can enforce temporary interruption of the surface water intake. **Groundwater quality** is more stable, and therefore less vulnerable for incidental changes. However, incidents can cause a permanent change of groundwater quality. It depends on the **resilience and reliability of the treatment** whether sudden variations in raw water quality can be handled well. | **Monitoring and evaluation** is necessary to be able to timely respond to a changing **water quality.** | To handle a varying or deteriorating **raw water quality** the **resilience and reliability of the drinking water treatment** must be extended. This may require innovations in treatment, which can lead to large investments, and higher **energy use** and an increase in **environmental impact of the treatment**. This may lead to a higher **drinking water tariff.** | |

## Appendix C  Results of analysis case 3 "Drinking water demand growth"

**Table C.1** *Summary of impact, short-term and long-term response and sustainability aspects in case 3, "Drinking water demand growth" (for complete results of the case study see Table C.2).*

| Impact | Short-term response | Long-term response | Sustainability aspects |
|---|---|---|---|
| A limited water resource availability will affect the drinking water availability. | See Table A.2. | See Table A.2. | Water resource availability, drinking water availability, resilience of drinking water supply, drinking water demand, water legislation. |
| A water quality deterioration affects the resilience and reliability of the drinking water treatment. | See Table B.2. | See Table B.2. | Water quality, drinking water treatment, reliability of treatment, drinking water standards. |
| A growing drinking water demand will put the reliability and resilience of the technical infrastructure under pressure. | See Table A.2. | Drinking water suppliers must adapt the technical infrastructure to the growing water demand. Water saving strategies may reduce the growth rate, which will limit the required extension of the technical infrastructure. | Drinking water demand, reliability of technical infrastructure, drinking water suppliers, drinking water availability, treatment, energy use, environmental impact, drinking water tariff. |
| A declining drinking water demand may also put the resilience of the technical infrastructure under pressure. | Research on potential risks of a decline in drinking water demand. | Adaptation strategies that increase the resilience of the infrastructure to growth as well as a decline of the drinking water demand. | Drinking water demand, reliability and resilience of technical infrastructure. |

*Table C.2* *Results of analysis of Case 3 "Drinking water demand growth", where additional to the analysis of the first two cases. The (additional) sustainability aspects in this case are displayed in bold. The grey cells refer to Table C.1.*

| Drivers | Pressure | Impact | Short-term response | Long-term response | Sustainability aspects |
|---|---|---|---|---|---|
| Changing climate variability, population growth, industrial developments | Limited water resource availability due to extreme weather events, other water use or limited abstraction permits | A limited water resource availability will affect the drinking water availability. | See Table A.2. | See Table A.2. | Water resource availability, drinking water availability, resilience of drinking water supply, drinking water demand, water legislation. |
| | | A limited **water resource availability** will affect the **drinking water availability**. The abstraction permits may be insufficient to meet the drinking demand, and possibilities to extend the permits will be minimal. This will put the **resilience of drinking water supply** to respond to changes in **drinking water demand** under pressure. This may cause frequent exceedance of permit conditions, or failure to the drinking **water legislation.** | See Table A.2. | See Table A.2. | |
| Changing climate variability, population growth, industrial developments | Surface water and groundwater quality deterioration | A water quality deterioration affects the resilience and reliability of the drinking water treatment. | See Table B.2. | See Table B.2. | Water quality, drinking water treatment, reliability of treatment, drinking water standards. |
| | | If the **water quality** deteriorates, this will affect the raw water quality of the water abstracted for drinking water production. The available **drinking water treatment** facilities may not be resilient to these changes. This affects the **reliability of the water treatment,** potentially causing exceedance of **drinking water standards.** | See Table B.2. | See Table B.2. | |
| Changing climate variability, population growth, | Growing drinking water demand | A growing drinking water demand will put the reliability and resilience of the technical infrastructure under pressure. | See Table A.2. | Drinking water suppliers must adapt the technical infrastructure to the growing water demand. Water saving strategies may reduce the growth rate, which will limit the required extension of the technical infrastructure. | Drinking water demand, reliability of technical infrastructure, drinking water suppliers, drinking water availability, |

| Drivers | Pressure | Impact | Short-term response | Long-term response | Sustainability aspects |
|---|---|---|---|---|---|
| industrial developments | | The overall capacity of the technical infrastructure determines whether the supply is resilient to respond to a higher **drinking water demand**. The drought in 2018 displayed technical limitations in parts of the drinking water supply system, putting the **reliability of the technical infrastructure** under pressure | See Table A.2 | Depending on the effectiveness of the **water saving** strategies that are developed, the technical limitations must be solved to meet the growing **drinking water demand**. **Drinking water suppliers** must solve the local aspects to ensure the **drinking water availability.** Because these adjustments take time, drinking water suppliers must start solving the aspects now. This requires substantial investments and also lead to an increasing **energy use** and **environmental impact,** which may result in an increasing **drinking water tariff.** | treatment, energy use, environmental impact, drinking water tariffs. |
| Socio-economic developments | Decrease in drinking water demand | A declining drinking water demand may also put the resilience of the technical infrastructure under pressure. | Research on potential risks of a decline in drinking water demand. | Adaptation strategies that increase the resilience of the infrastructure to growth as well as a decline of the drinking water demand. | Drinking water demand, reliability and resilience of technical infrastructure. |
| | | If at some moment the socio-economic developments reverse the **drinking water demand** growth, the **reliability and resilience of the technical infrastructure** will be put under pressure. Especially when the focus is on dealing with a growing **water demand**, there is the risk of over-dimensioning of the technical infrastructure. This will put the **drinking water quality** under pressure in case of a decreasing **drinking water demand**. | While working on solutions for the growing **drinking water demand**, it is important to consider the potential risks of a decreasing demand. | The chosen adaptation strategies for a growing drinking water demand must also be resilient and reliable under a decreasing drinking water demand. | |

# Appendix D  Summary of Sustainable Development Goal 6 targets and indicators related to sustainability characteristics

*Table D.1* Summary Sustainable Development Goal 6 targets and indicators related to sustainability characteristics

| Target | Indicator | Hydrological system | | | Technical system | | | Socio-economic system | | |
|---|---|---|---|---|---|---|---|---|---|---|
| | | Water quality | Water resource availability | Impact of drinking water abstraction | Reliability of technical infrastructure | Resilience of technical infrastructure | Energy use and environmental impact | Drinking water availability | Water governance | Land and water use |
| 6.1 By 2030, achieve universal and equitable access to safe and affordable drinking water for all | 6.1.1 Proportion of population using safely managed drinking water services | | | | x | x | | x | | |
| 6.2 By 2030, achieve access to adequate and equitable sanitation and hygiene for all and end open defecation, paying special attention to the needs of women and girls and those in vulnerable situations | 6.2.1 Proportion of population using safely managed sanitation services, including a hand-washing facility with soap and water | | | | | | | | | |
| 6.3 By 2030, improve water quality by reducing pollution, eliminating dumping and minimizing release of hazardous chemicals and materials, halving the proportion of untreated wastewater and substantially increasing recycling and safe reuse globally | 6.3.1 Proportion of wastewater safely treated<br>6.3.2 Proportion of bodies of water with good ambient water quality | x<br>x | | | | | | | x<br>x | x<br>x |
| 6.4 By 2030, substantially increase water-use efficiency across all sectors and ensure sustainable withdrawals and supply of freshwater to address water scarcity and substantially reduce the number of people suffering from water scarcity | 6.4.1 Change in water-use efficiency over time<br>6.4.2 Level of water stress: freshwater withdrawal as a proportion of available freshwater resources | | x | x | x<br>x | x | x<br>x | | x<br>x | x<br>x |
| 6.5 By 2030, implement integrated water resources management at all levels, including through transboundary cooperation as appropriate | 6.5.1 Degree of integrated water resources management implementation (0–100)<br>6.5.2 Proportion of transboundary basin area with an operational arrangement for water cooperation | x | x | | | | | | x<br>x | |
| 6.6 By 2020, protect and restore water-related ecosystems, including mountains, forests, wetlands, rivers, aquifers and lakes | 6.6.1 Change in the extent of water-related ecosystems over time | | | x | | | | | x | |

| Target | Indicator | Hydrological system | | | Technical system | | | Socio-economic system | | |
|---|---|---|---|---|---|---|---|---|---|---|
| | | Water quality | Water resource availability | Impact of drinking water abstraction | Reliability of technical infrastructure | Resilience of technical infrastructure | Energy use and environmental impact | Drinking water availability | Water governance | Land and water use |
| 6.a By 2030, expand international cooperation and capacity-building support to developing countries in water and sanitation-related activities and programmes, including water harvesting, desalination, water efficiency, wastewater treatment, recycling and reuse technologies | 6.a.1 Amount of water- and sanitation-related official development assistance that is part of a government coordinated spending plan | | | | | | | | x | |
| 6.b Support and strengthen the participation of local communities in improving water and sanitation management | 6.b.1 Proportion of local administrative units with established and operational policies and procedures for participation of local communities in water and sanitation management | | | | | | | | x | |

# Appendix E    Overview of sustainability characteristics and criteria

Table E.1 summarises the hydrological, technical and socio-economic sustainability characteristics and criteria for a local drinking water supply system from Section 3.[2]

| System | Sustainability characteristics | Sustainability criteria | General description | Sustainable | Under pressure | Unsustainable | Suggestions for general data sources | Reference for general data sources |
|---|---|---|---|---|---|---|---|---|
| **Hydrological system** | Water quality | Current raw water quality | To which extent does the current raw water quality meet set standards? | Current raw water quality meets set standards | Occasionally the current raw water quality exceeds set standards | Current raw water quality is permanently exceeding set standards | e.g. Status of water bodies according to European Water Framework Directive | European Union (2000) |
| | | Chemical aspects of water quality | Which trends are found in chemica water quality development? | Chemical water quality is improving | Consistent chemical water quality | Deteriorating chemical water quality | Ibid | Ibid |
| | | Microbial aspects of water quality | To which extent is microbial pollution a threat to the raw water quality? | No risk of microbial pollution | Microbial pollution is a potential risk, but the microbiological quality is sufficient | Microbial pollution is an actual risk and the microbiological quality is insufficient | Ibid | Ibid |
| | | Acceptability aspects of water quality | Are there aspects of water quality that limit the acceptability of the drinking water (salinization, hardness, colour)? | No issues with acceptability of the drinking water | Salinization, hardness or colour cause a minor acceptability issue | Salinization, hardness and/or colour cause serious acceptability issues | Ibid | Ibid |
| | | Monitoring and evaluation of water quality trends | Is there sufficient and adequate monitoring and evaluation of water quality trends available? | Sufficient and adequate monitoring and evaluation of water quality trends | There is monitoring available, but evaluation of data is limited, resulting in a limited understanding of water quality trends | There is limited or no monitoring available, and water quality trends are not investigated | Ibid | Ibid |
| | Water resource availability | Surface water quantity | Are there current limitations or future threats to the abstracted surface water volume? | Sufficient availability all year round or 'no surface water abstraction' | Surface water availability varies during the year and may occasionally be limited in case of dry weather conditions | There is regularly insufficient surface water volume available in the dry season | e.g. Status of water bodies according to European Water Framework Directive | European Union (2000) |

[2] This appendix is an extended and updated version of appendix A of Van Engelenburg e.a. 2019

| System | Sustainability characteristics | Sustainability criteria | General description | Sustainable | Under pressure | Unsustainable | Suggestions for general data sources | Reference for general data sources |
|---|---|---|---|---|---|---|---|---|
| | | Groundwater quantity | Are there current limitations or future threats to the abstracted groundwater volume? | Abstraction is not limited because groundwater is recharged sufficiently (yearly abstraction<annual recharge minus environmental streamflow) or 'no groundwater abstraction' | Abstraction is not limited but exceeds annual recharge minus environmental streamflow | Abstraction volume is limited because groundwater is abstracted from a confined aquifer that is not recharged ('mining') | e.g. Status of water bodies according to European Water Framework Directive | European Union (2000) |
| | | Other available water resources | Are there water resources available for drinking water production other than currently used? | There are sufficient water resources available that could replace the current used water resource with minor adjustments to the drinking water treatment method | There are other water resources available that could replace the current used water resource, but this will require major adjustments to the drinking water treatment method | There are no water resources available that could replace the current used water resource | e.g. Status of water bodies according to European Water Framework Directive | European Union (2000) |
| | | Vulnerability used water system for contamination | To which extent is the used water system vulnerable for contamination? | The water system is hardly vulnerable for contamination because the used water resource is protected by an aquitard (groundwater in confined aquifers) | The water system is vulnerable for soil and groundwater pollution (phreatic groundwater) | The water system is vulnerable for calamities and diffuse contamination (surface water) | e.g. Status of water bodies according to European Water Framework Directive | European Union (2000) |
| | | Natural hazards and emergencies risk | To which extent are natural hazards (droughts, floods, earthquakes, forest fires) threatening the water resources availability? | Limited risk of natural hazards (<1 per 25 years) | Minor risk of a natural hazard (< 1 per 10 years) | Natural hazards occur frequently (> 1 per 10 years) and are a serious threat to water resources availability | e.g. National flood risk inventory, CSD Indicator of Sustainable Development (Percentage of population living in hazard prone areas) | UN (2007) |
| | Impact of drinking water abstraction | Impact to surface water system | The scale of impact of the abstraction to the surface water system | Small (groundwater abstraction below aquitard) | Medium (riverbank abstraction, phreatic groundwater abstraction) | Large (surface water abstraction) | e.g. Status of water bodies according to European Water Framework Directive | European Union (2000) |

| System | Sustainability characteristics | Sustainability criteria | General description | Sustainable | Under pressure | Unsustainable | Suggestions for general data sources | Reference for general data sources |
|---|---|---|---|---|---|---|---|---|
| | | Impact to groundwater system | The scale of impact of the abstraction to the groundwater system | Small (surface water abstraction) | Medium (riverbank abstraction, groundwater abstraction below aquitard) | Large (phreatic groundwater abstraction) | e.g. Groundwater footprint | Gleeson and Wada (2013) |
| | | Balance between annual recharge and abstraction | The balance between abstraction and recharge of the water system | The net abstraction volume is less than 10 % of the average annual recharge in the recharge area | The net abstraction volume is 10-40 % of the average annual recharge in the recharge area | The net abstraction volume is > 40 % of the average annual recharge in the recharge area | SSI (Renewable water resources) | Van der Kerk and Manuel (2008) |
| | | Hydrological compensation | The extent to which the impact of abstraction is compensated hydrologically | Small impact or impact is hydrologically compensated with a technical measure | There are possibilities for hydrological compensation of the impact of the abstraction, but they are not operational yet | There is a significant impact of the abstraction, but there are no possibilities for hydrological compensation | Local hydrological knowledge, hydrological modelling results | e.g. Van Engelenburg et al. (2017), Van Engelenburg et al. (2020a) |
| | | Spatial impact of abstraction facility/ storage/reservoir | Size of required working area for abstraction facility | Small (groundwater abstraction with basic treatment facility) | Medium (groundwater abstraction with medium treatment facility) | Large (surface water abstraction with storage basins and extended treatment facility) | Drinking water company's information, map | |
| **Technical system** | Reliability of technical infrastructure | Technical state abstraction and treatment facility | Is the technical state of the drinking water production facility sufficient and fully deployable? | The technical state of the drinking water production facility is sufficient and fully deployable | Production capacity is sufficient but not fully deployable due to restrictions in permit or technical limitations | Production capacity is insufficient due to technical limitations | IWA (Ph1 Treatment plant utilisation) | Alegre et al. (2006b) |
| | | Technical state distribution infrastructure | Are there issues that complicate the drinking water distribution? | The distribution infrastructure is adequate to meet the required distribution capacity and water pressure | The distribution infrastructure is in general adequate but at extreme peak demand limitations in the drinking water distribution cause reduced water pressure and limited drinking water supply | The distribution infrastructure is insufficient and major disruptions of the drinking water supply occur regularly | Performance data of water utilities | e.g. Dutch Drinking Water Law (2009) |

| System | Sustainability characteristics | Sustainability criteria | General description | Sustainable | Under pressure | Unsustainable | Suggestions for general data sources | Reference for general data sources |
|---|---|---|---|---|---|---|---|---|
| | | Complexity of water treatment | How complex is the required treatment and is the treatment effective to meet the water quality issues? | Technical water quality issues (iron/manganese removal, pH-correction), require only basic treatment | Water quality issues such as hardness require medium complex treatment (decalcification) | Serious water quality issues (chemical, microbiological) require a complex treatment (ultra-filtration, reversed osmosis) | Performance data of water utilities | e.g. Dutch Drinking Water Law (2009) |
| | | Supply continuity for customers | Are there frequent drinking water supply interruptions? | Drinking water supply interruptions < 1 hr per year | Drinking water supply interruptions < 10 days per year | Drinking water supply interruptions > 10 days per year | Performance data of water utilities, IWA (QS17 Days with restrictions to water service) | Alegre et al. (2006b) |
| | | Operational reliability | Is the facility operationally reliable? | Facility meets corporate standard for operational reliability | The facility does not fully meet corporate standard for operational reliability, but investments are planned to increase the operational reliability < 5 years | Facility is not operationally reliable and there are no investments planned to improve the reliability within 5 years | Performance data of water utilities | e.g. Dutch Drinking Water Law (2009) |
| | | Abstraction permit compared to annual drinking water demand | Are the permitted abstraction volumes sufficient to meet the annual drinking water demand? | The permitted abstraction volumes are sufficient to meet the current and future annual drinking water demand (operational reserve > 10%) | The permitted abstraction volumes are sufficient to meet the current annual drinking water demand but cannot meet the future demand (operational reserve < 10%) | The permitted abstraction volumes are insufficient to meet the current of future annual drinking water demand | Performance data of water utilities | e.g. Dutch Decree on Water (2007) |
| | Resilience of technical infrastructure | Production capacity compared to peak demand | Is the production capacity per hour sufficient to meet extreme peak demand? | The production capacity per hour is sufficient to meet extreme peak demand | The production capacity is < 5% below the predicted extreme peak demand and therefore is not fully sufficient | The production capacity is > 5% below the predicted extreme peak demand and therefore is insufficient to meet peak demand | Performance data of water utilities, IWA (Ph1 Treatment Plant Utilisation) | Alegre et al. (2006b) |
| | | Flexibility of treatment method for changing raw water quality | Is the treatment method flexible to respond to a changing raw water quality? | The treatment method removes a broad spectrum of pollutants and therefore can also handle various new pollutants (e.g. membrane treatment methods) | The treatment method is flexible when concentrations of the currently removed elements change, but cannot remove other pollutants (e.g. decalcification) | The treatment method is not flexible to respond to large changes in concentrations or pollutants (e.g. sand filtration) | Performance data of water utilities | |

| System | Sustainability characteristics | Sustainability criteria | General description | Sustainable | Under pressure | Unsustainable | Suggestions for general data sources | Reference for general data sources |
|---|---|---|---|---|---|---|---|---|
| | | Technical innovations to improve resilience | Are technical innovations developed to improve resilience? | Within society there is an ongoing research to find technical innovations on drinking water use or supply to improve resilience | Within the drinking water company there is an ongoing research to find technical innovations for drinking water supply to improve resilience | There is no or limited research on technical innovations for drinking water supply | Data of water utilities (annual report) | |
| | | Technical investments to improve resilience | Are technical investments made to improve resilience? | Technical investments are made to improve the resilience of the drinking water infrastructure, including investments in technical innovations | There is a limited budget for technical investments to improve the resilience of the drinking water infrastructure | There is no budget for technical investments. | Financial data of water utilities | |
| | | Energy use of abstraction and treatment | Energy use for abstraction and treatment of water per m3 | Low (shallow groundwater abstraction, short distance to treatment, basic treatment) | Average (deep groundwater abstraction, short distance to treatment, medium treatment groundwater) | High (long transport distance to treatment, complex treatment) | IWA Ph5 Standardised energy consumption | Alegre et al. (2006b) |
| | | Energy use of distribution | Energy use for distribution | Low (average transport distances < 15 km) | Average (average transport distances < 30 km) | High (average transport distances > 30 km) | EBC (electricity use) | European Benchmarking Co-operation (2017) |
| | Energy use and environmental impact | Environmental impact (additional excipients, waste-water, waste materials) | Are there materials used or produced in the treatment with an environmental impact? | No use or produce of materials with high environmental impact | Use of additional excipients with high environmental impact in the treatment | Production of waste materials and wastewater with high environmental impact | EBC (climate footprint) | European Benchmarking Co-operation (2017) |
| | | Reliability energy supply | Is the energy supply reliable? | Reliable energy supply and emergency energy backup | Average reliable energy supply, no emergency energy backup | Unreliable energy supply, no emergency energy backup | EBC (electricity use) | |
| | | Use of renewable energy | Use of renewable energy sources (own generation or acquired green energy) | All used energy is renewable energy | > 50 % renewable energy is used | < 50 % renewable energy | IWA Ph7 Energy recovery | Alegre et al. (2006b) |
| **Socio-economic system** | Drinking water availability | Percentage connected households | Households directly connected to drinking water supply system | > 95 % | 80 - 95 % | < 80 % | IWA QS3 Population coverage | Alegre et al. (2006b) |
| | | Drinking water service quality | Continuity and quality of supply (local scale) | Continuity and quality of drinking water supply guaranteed 24/7 | Continuity of drinking water supply or quality under pressure at peak demand | Drinking water quality and supply continuity not guaranteed | IWA QS12 Continuity of supply, QS18 Quality of supplied water | Alegre et al. (2006b) |

| System | Sustainability characteristics | Sustainability criteria | General description | Sustainable | Under pressure | Unsustainable | Suggestions for general data sources | Reference for general data sources |
|---|---|---|---|---|---|---|---|---|
| | | Drinking water tariff | Average water charges without public charges (company scale) | < 1 €/m3 | 1 - 2 €/m3 | > 2 €/m3 | IWA Fi28 Average water charges for direct consumption | Alegre et al. (2006b) |
| | | Water saving strategy | Water saving strategy to reduce average water demand in litre per person per day (national scale) | Effective water saving strategy resulting in an average water demand < 100 l pp pd | Water saving strategy aiming to reduce the average water demand of 100-200 l pp pd | No water saving strategy | SSI (Sufficient to drink) | Van der Kerk and Manuel (2008) |
| | | Water safety protocols | Are there water safety protocols or water safety plans to safeguard the drinking water supply? | Water safety protocols fully cover the drinking water supply and the organisation is performing accordingly | There are safety protocols, but only covering a part of the drinking water supply of not fully performed | There are no safety protocols | Drinking water company's information | e.g. Dutch Drinking Water Law (2009) |
| | | Availability of (drinking) water legislation and policies | Is there adequate legislation on drinking water supply and is there enforcement of this legislation? | There is adequate legislation on drinking water supply combined with sufficient enforcement by legal authorities | There is legislation on drinking water supply but limited or no enforcement by legal authorities | There is no legislation and enforcement on drinking water supply | SSI (Good Governance), national and local legislation | Van der Kerk and Manuel (2008) |
| | | Compliance of drinking water supplier | Are the required permits available, and is the facility compliant to the permit requirements? | All permits are available, and the facility is compliant to the permit requirements | The permits are available, but the facility is not fully compliant to the permit requirements | There is a lack of adequate drinking water supply legislation and drinking water suppliers only follow their company's standard | SSI (Good Governance), permits, TRUST Framework for UWCS-sustainability (G1-G4) | Van der Kerk and Manuel (2008); Brattebø et al. (2013) |
| | Water governance | Decision-making process by (local) authorities | Are local stakeholders involved in decisions on drinking water supply or the water system? | Local stakeholders are involved in the planning process and can participate in licensing procedures | Local stakeholders are not involved in the planning process but cannot participate in licensing procedures | Local stakeholders cannot easily involve in the decision-making process | SDG 6.b | UN (2015) |
| | | Local stakeholder interests | Does the local authority actively weigh stakeholder interests in the decision-making process? | Stakeholders are involved in the decision-making process and stakeholder interests must be taken into account in the licensing process legally | Stakeholder interests must be taken into account in in the licensing process | The interests of (some) local stakeholders are not accounted for by the local authorities | SDG 6.b, national or local legislation | UN, 2015 |
| | | Emergency risk caused by human activities or conflicts | Is there emergency risk caused by human activities or conflicts? | There is in general no serious emergency risk caused by human activities or conflicts | There is a low emergency risk caused by human activities | There is an evident emergency risk caused by human activities or conflicts | SDG 16 | UN, 2015 |

| System | Sustainability characteristics | Sustainability criteria | General description | Sustainable | Under pressure | Unsustainable | Suggestions for general data sources | Reference for general data sources |
|---|---|---|---|---|---|---|---|---|
| | | Land use (including subsurface use) | Is land or subsurface use in the area posing a threat to the drinking water supply? | The impact of land or subsurface use is limited due to low-risk use or because the drinking water supply is well protected against the impact | The land or subsurface use forms a potential risk to the drinking water supply but is regulated | The land or subsurface use is affecting the drinking water supply | e.g. Status of water bodies according to European Water Framework Directive | European Union (2000) |
| | | Water use for other purposes than drinking water | Does water use in the area pose a threat to the drinking water supply? | In general, there is sufficient water available for all functions and water quality is not affected by water use | In extreme situations the available water resources are limited and must be fairly distributed between water users, or water quality deteriorates | There is constantly insufficient water available for all water users and/or water quality deterioration due to various water use | e.g. Status of water bodies according to European Water Framework Directive | European Union (2000) |
| | Land and water use | Regulations on land and water use | Are there regulations on land use and underground activities to protect the local drinking water abstraction? | There are regulations to remove unwanted activities from the recharge area to protect the local drinking water abstraction | There are regulations to prevent new unwanted activities by using the stand-still/step forward principle | There are no regulations to protect the local drinking water abstraction | (Inter)national legislation, TRUST Framework for UWCS-sustainability (G1-G4) | e.g. Dutch Decree on Water (2007), Brattebø et al. (2013) |
| | | Limitations to land or water use | Is the presence of the facility a significant impediment for current or future land use of underground activities? | The drinking water supply does not present a significant impediment for land or subsurface use | The drinking water supply limits future land use or underground activities | The drinking water supply is a significant impediment for current as well as future land use or underground activities | e.g. Status of water bodies according to European Water Framework Directive | European Union (2000) |
| | | Financial compensation of economic damage from impact of abstraction or limitations to land use | Is there financial compensation of economic damage from the impact of abstraction or limitations to land use? | Financial compensation of economic damage caused by the drinking water supply is organised based on legislation | Drinking water suppliers financially compensate economic damage based on bilateral agreements | There is financial compensation of economic damage caused by the drinking water supply company | National or local legislation | e.g. Dutch Decree on Water (2007) |