# Peer review of "Sustainability characteristics of drinking water supply in the Netherlands"

_Drinking Water Engineering and Science, 2020_

## Referee Comment (RC1) · Anonymous Referee #1 · 21 Apr 2020

Summary: The authors aim to present a systems approach to an integrated assessment of drinking water supply based on narratives of three case studies using DPSIR approach.

Comments: being a quantitative researcher with exposure to narrative style research, I could not appreciate the content presented. The authors allude to an integrated assessment based on system thinking for the first time but only stick to DPSIR framework without motivating its choice. There were mentions of socioecological and sociotechnical systems but I didnt see much content coming out from those respective disciplines, except perhaps DPSIR to certain extent. Why didnt the authors think of system dynamics models that explicitly incorporate feedbacks and are capable of integrating fast and slow dynamical systems. This also then extends to the way case studies were

dealt with. Given that DPSIR approach is rather linear, I found key important aspects of feedbacks, synergies and tradeoffs between various driving, state, impact and response variables. For example, some pressures such as due to population growth might be influenced by policy responses of past actions such as providing reliable and abundant water supply. These are quite important if SDGs are to be investigated. In this regard I found the choice of the framework used by the authors as not well justified. I also had difficulties appreciating the discussion as I found tables synthesizing aspects of the three case studies repetitive.

If the authors are intending to revise and resubmit, I would challenge the authors on providing a more sound basis for the choice of DPSIR framework in their pursuit of holistically assessing the sustainability of drinking water supply systems while not ignoring key aspects of feedbacks between slow and fast dynamics of sociohydrological systems that supply systems are embedded in. What could have been innovative would perhaps be a narrative treatment of how water supply systems might themselves have emerged from the underlying sociohydrological dynamics, locking them into a path towards unsustainable development (e.g. water supply systems that emerged in water abundant/flood prone countries might not be as resilient to drought events as those that emerged in latter drought prone systems). The case studies presented provide abundant material to shift the narrative in this direction.

---

## Referee Comment (RC2) · Anonymous Referee #2 · 24 Jul 2020

Sustainability characteristics of water supply were determined based on the analysis of three case studies in the Netherlands. From there general sustainability criteria were identified that can be used in to assess drinking water supply. The paper tackles an interesting subject and is well written. However, it need some adjustment before publication General comments: - The title should include the fact that the study was based on three case studies in the Netherlands - The paper is rather long, the writing could be more concised and redundancies should be eliminated. - When general statements are done, they should be supported by literature - The methodology chapter is rather general, without a good description how sustainability characteristics and criteria were precisely determined. - In addition, the sustainability criteria should be better formulated in order to be able to judge compliance (or not) - When looking at the general

use of the criteria for judgement of water supply systems in the world, at least some criteria are missing, such as Non-revenue water/leakage (which is maybe not a question in the Netherlands, but internationally it is) for technical system; and cost-recovery, a good-billing system, transparency in water tariffs, equality in water billing, etc. for governance. These flaws can may be avoided by better (and more systematically) addressing previous bullets. - Description of cases should be part of methodology section. - Avoid repetition of results in the various tables. It is better to summarize at once and then describe in the various sections. - Discussion (with literature) should be part of "results" section and not of "conclusions" and conclusions should be concised. - Language, including tenses, should be checked: present tense for general statements and past tense for own findings and work. - Avoid word "issues", but better "characteristics" "criteria" "aspects", depending on own definition. Specific comments: - Line 40-44: delete (see general comments) - Line 48: delete and give summary of results - Line 56-57: too general, delete - Line 57-60: give reference - Line 64-67: could be shortened (little information), only references sufficient - Line 70-71: delete sentence - Line 84: delete sentence - Line 90-96: not much extra information (too general without references), so consider deleting. - Line 102-104: delete sentence - Line 127-130: not relevant information - Line 131: internal colleagues = staff - Line 135: how the authors came to the defined "sustainability characteristics"? - Line 142: can be = could potentially be - Line 144: Figure 1 does not give much extra information in relation to text so can be deleted. - Line 147 and onwards: Section 2.1 is too general with a few references. Could be shortened to in or two sentences as introduction. - Line 178 and onwards: could be more concised too, by at least deleting 178-182 - Line 210: Figure gives little extra information, so could be deleted. By the way, when it is not an own figure, a reference should be given. - Line 214: case selection should be more to the point - Line 215-219: general information without references, could be deleted. - Line 233-240: avoid redundant information (already explained elsewhere) - Line 246 and onwards: use Italic for the DPSIR elements - Line 246 and onwards: give references for the information that is given on the cases (e.g. line 247, 257, 259, 260-263, . . ..)

- Line 260-263: how this information is obtained/verified? - Line 274 and onwards: Is there a "case" or is it a "general" description. Now that is not clear.. Better, focus on the "Vitens case". - Line 343-352: redundant information, so delete. - Line 360: introduce JPM - Line 367: check table 4, e.g. what is difference between "raw water" and "surface water" or "groundwater"; "impact of abstraction" is redundant; "water quantity" = "water flows". See also general comments. - Line 378-383: too speculative. Please stick to own findings (and discuss in relation to literature). - Line 390: this will also impact costs of investments and thus water tariff. - Line 396: delete sentence - Line 444 and 469: why is the existence of a WSP a sustainability criterium? - Line 476 and onwards: avoid repetitions with previous sections, see earlier comments. - Line 490-501: delete (see general comments) - Line 503-510: delete (see general comments) - Line 521-529: delete (see general comments)

---

## Author Comment (AC1) · 28 Aug 2020

Dear editorial board, referees,

Herewith we respond to the reviews of the anonymous referees. We hope the attached rebuttal will provide you a clear overview of our response and the adjustments we propose to make to the original manuscript to meet the referees' remarks.

Kind regards,

The authors: Jolijn van Engelenburg Erik van Slobbe Adriaan J. Teuling Remko Uijlenhoet Petra Hellegers

Please also note the supplement to this comment:

[Figure]

https://dwes.copernicus.org/preprints/dwes-2020-8/dwes-2020-8-AC1-supplement.pdf

[Figure]

**Supplement:**

**Rebuttal**

| Journal Name: | Drinking Water Engineering and Science |
|---|---|
| Manuscript Number or original submission: | Dwes-2020-8 |
| Title of the original Manuscript: | Sustainability characteristics of drinking water supply |
| New Title of the Revised Manuscript | Sustainability characteristics of drinking water supply in the Netherlands |
| Type of the Article | Research article |
| Keywords old manuscript | Systems approach; DPSIR; drinking water supply; local scale; sustainability |
| New Keywords of the Revised Manuscript | Systems approach; drinking water supply; local scale; sustainability; the Netherlands |

*Dear editorial board,*

*Herewith we respond to the reviews of the anonymous referees. We hope this rebuttal will provide you a clear overview of our response and the adjustments we propose to make to the original manuscript to meet the referees' remarks.*

*Kind regards,*
*The authors*

| # | Referee #1 Comments () | Authors Comments | Adjustments in new manuscript |
|---|---|---|---|
| 1 | Comments: being a quantitative researcher with exposure to narrative style research, I could not appreciate the content presented. | This paper is part of an interdisciplinary research project on drinking water supply, performed by researchers both with quantitative as well as more qualitative disciplinary backgrounds (international water resources management, hydrology, climate change studies, drinking water supply) aiming to contribute to the sustainability of drinking water supply. Additional to the more quantitative research on the hydrological impact of drinking water supply, the researchers also were confronted with the complexity of research on sustainable drinking water supply strongly manifested. This urged us to use a systems approach that allowed combining quantitative and qualitative characteristics. We propose to do this by identifying the most relevant challenges that must be addressed in policy development on | In the final paragraph section 1 we will add to the aim of the research: This research aims to propose a set of sustainability characteristics that describe the drinking water supply system on a local scale *to support policy- and decision-making on sustainable drinking water supply.* |

| # | Referee #1 Comments () | Authors Comments | Adjustments in new manuscript |
|---|---|---|---|
| | | sustainable drinking water supply, offering policy makers and planners an evidence based approach for assessing sustainability of drinking water supply from their perspective. | |
| 2 | The authors allude to an integrated assessment based on system thinking for the first time but only stick to DPSIR framework without motivating its choice. There were mentions of socioecological and sociotechnical systems but I didnt see much content coming out from those respective disciplines, except perhaps DPSIR to certain extent. Why didnt the authors think of system dynamics models that explicitly incorporate feedbacks and are capable of integrating fast and slow dynamical systems. This also then extends to the way case studies were dealt with. Given that DPSIR approach is rather linear, I found key important aspects of feedbacks, synergies and tradeoffs between various driving, state, impact and response variables. For example, some pressures such as due to population growth might be influenced by policy responses of past actions such as providing reliable and abundant water supply. These are quite important if SDGs are to be investigated. In this regard I found the choice of the framework used by the authors as not well justified. | To reach the aim of this research to support policy development on sustainable drinking water supply, we chose to analyse 3 practice cases to identify the main sustainability aspects in these cases. For this we decided to use DPSIR. DPSIR has previously been used for complex water systems by various well-known researchers in this field, such as Claudia Pahl-Wostl. In Binder, Hinkel et al. (2013) a comparison was made between various frameworks. The authors of that paper concluded that DPSIR was a policy framework that does not explicitly include development of a model, but aims at providing policy relevant information, on pressures and responses on different scales. In Carr, Wingard et al. (2009) the use of DPSIR for sustainable development was evaluated. Although the authors were critical regarding the use of the DPSIR framework on national, regional or global scales, they considered application on a local scale appropriate. They concluded that practitioners can use DPSIR for local-scale studies because it assesses the place-specific nuances of multiple concerned stakeholders more realistically. In Van Noordwijk, Speelman et al. (2020) DPSIR was used to understand the joint multi-scale phenomena in the forest-water-people nexus and thus diagnosed issues to be addressed in serious games for local decision-making. | This will be elaborated in the (new) section 2.1 on the case analysis method. |
| | | In the discussion we reflected on the limitations of the linear DPSIR approach with regard to the trade-offs and feedbacks in the drinking water supply. While the aim of the research was to identify sustainability characteristics for drinking water supply on a local scale to support policy development and stakeholder involvement rather than analysis and modeling of the system dynamics, we decided to use this framework. A next step could potentially be to use the identified system characteristics for a system dynamics analysis and modeling. However, this is beyond the scope of this current research. | In the new discussion section 4.1 this will be elaborated further. |
| 3 | I also had difficulties appreciating the discussion as I found tables synthesizing aspects of the three case studies repetitive. If the authors are intending to revise and resubmit, I would challenge the authors on providing a more sound basis for the choice of DPSIR framework in their pursuit of holistically assessing the sustainability of drinking water supply systems while not ignoring key aspects of feedbacks between slow and fast dynamics of | The tables of the case studies indeed show repetitive issues. This will be solved by removing the summarizing tables 1-3 and referring to the adjusted Appendices A-C. Concerning the remark on the dynamics in the sociohydrological, as well as the sociotechnical dynamics, we refer to the aim of the research, which was to identify the most relevant challenges that must be addressed in policy development on sustainable drinking water supply, rather than the system dynamics. In our discussion we did address the fact that the feedbacks and trade-offs in the drinking water supply cases complicated the DPSIR | App. A-C (adjusted/new)

See above.

Section 4.1 (discussion on use of DPSIR) |

| # | Referee #1 Comments () | Authors Comments | Adjustments in new manuscript |
|---|---|---|---|
| | sociohydrological systems that supply systems are embedded in. | analysis. However, for the aim of the research, the DPSIR approach sufficed. Use of a different integrated systems approach would not have led to a significantly different outcome of the research. | |
| 4 | What could have been innovative would perhaps be a narrative treatment of how water supply systems might themselves have emerged from the underlying sociohydrological dynamics, locking them into a path towards unsustainable development (e.g. water supply systems that emerged in water abundant/flood prone countries might not be as resilient to drought events as those that emerged in latter drought prone systems). The case studies presented provide abundant material to shift the narrative in this direction. | Evaluation of how water supply systems developed as a result of underlying sociohydrological dynamics would indeed be a very interesting research topic. The case studies could definitely be used for this, when combined with case studies in semi-arid/arid countries. However, this is beyond the scope of the current study, which was to find sustainability characteristics. | No adjustment in manuscript |

| # | Referee #2 Comments () | Author's comments | Adjustments in new manuscript |
|---|---|---|---|
| | Sustainability characteristics of water supply were determined based on the analysis of three case studies in the Netherlands. From there general sustainability criteria were identified that can be used in to assess drinking water supply. The paper tackles an interesting subject and is well written. However, it need some adjustment before publication | Thank you for your kind words. | |
| | **General comments:** | | |
| 5 | - The title should include the fact that the study was based on three case studies in the Netherlands | Thank you for this suggestion. The title will be adjusted accordingly (adding "in The Netherlands"). | Title |
| 6 | - The paper is rather long, the writing could be more concised and redundancies should be eliminated. | We will remove several figures, integrate some of the tables and remove redundancies following your suggestions. | Fig 1-2, Table 1-3 are removed |
| 7 | - When general statements are done, they should be supported by literature | Where available references will be added. If not available the statement will be removed. | |
| 8 | - The methodology chapter is rather general, without a good description how sustainability characteristics and criteria were precisely determined. | The adopted research approach consisted of four steps. The first step was the selection and analysis of three drinking water practice cases in the Netherlands, aiming to identify the Dutch sustainability aspects in these cases. Three Dutch cases were selected based on their impact to the sustainability of drinking water supply in the Netherlands, illustrated with Vitens data (Van Engelenburg, Fleuren et al. 2020). In the second step the cases were analysed using the DPSIR framework (see section 2.1). The sustainability aspects of these cases were | The method is more precisely described and clarified in section 2. |

| | | identified in the descriptive results of the DPSIR analysis. The results were combined with Dutch governmental reports on these events and developments (Vitens 2016, Ministry of Infrastructure and Environment and Ministry of Economic Affairs and Climate Policy 2019) and cross-checked with Vitens staff. The sustainability aspects were categorized into hydrological, technical and socio-economic aspects. This resulted in a set of relevant sustainability aspects. The following step was used to broaden the perspective from the drinking water supply in the Netherlands to a more general perspective, by cross-checking the set of sustainability aspects with the targets and indicators in Sustainable Development Goal 6 (UN 2015), and the WHO Guidelines for Drinking-Water Quality (WHO 2017). Based on the analysis nine hydrological, technical and socio-economic sustainability characteristics were proposed that cover the identified sustainability aspects.
In the final step of the study each sustainability characteristic was elaborated further into five sustainability criteria that describe the local drinking water supply system.
This resulted in a proposal for sustainability characteristics and criteria of local drinking water supply systems that could potentially be applied in various contexts. | |
|---|---|---|---|
| 9 | - In addition, the sustainability criteria should be better formulated in order to be able to judge compliance (or not) | We will provide an additional appendix to the current paper that formulates and elaborates the sustainability criteria in the following information for each criteria: general explanation of the criterion, description of what may be considered sustainable, under pressure and unsustainable, and suggestions for indicators or other date sources.
. | Additional detailed information will be provided in Appendix E |
| 10 | - When looking at the general use of the criteria for judgement of water supply systems in the world, at least some criteria are missing, such as Non-revenue water/leakage (which is maybe not a question in the Netherlands, but internationally it is) for technical system; and cost-recovery, a good-billing system, transparency in water tariffs, equality in water billing, etc. for governance. These flaws can may be avoided by better (and more systematically) addressing previous bullets. | The mentioned criteria that the second referee found missing, are implicitly accounted for in the sustainability criteria. Non-revenue water/leakage in "Technical state distribution infrastructure", cost-recovery/billing system/tariffs are implicitly accounted for in the governance criteria "Availability of (drinking) water legislation and policies" and "Compliance of drinking water supplier". We will make this more clear in the elaboration of the sustainability criteria in the appendix as mentioned above. | Additional detailed information will be provided in Appendix E |
| 11 | - Description of cases should be part of methodology section. | The case description will be moved to the method section (section 2.2), and section 3 will be limited to the results of the analysis. | Section 2.2 and section 3 |
| 12 | - Avoid repetition of results in the various tables. It is better to summarize at once and then describe in the various sections. | We will integrate Table 1-3 into Appendix A. In addition we will move Table 7 towards a new Appendix and elaborate this into a further description of the criteria. | App. A-C, App. E. |
| 13 | - Discussion (with literature) should be part of "results" section | We will adjust the final section, separating the discussion from the final conclusion of the | Section 4 Discussion and |

| | | | |
|---|---|---|---|
| | and not of "conclusions" and conclusions should be concised. | paper in two separate sections, with specific attention to the conciseness of the writing. | Section 5 Conclusion |
| 14 | - Language, including tenses, should be checked: present tense for general statements and past tense for own findings and work. | This will be checked and adjusted in the revised manuscript. | |
| 15 | - Avoid word "issues", but better "characteristics" "criteria" "aspects", depending on own definition. | We will replace the word 'sustainability issue' to 'sustainability aspect'. These aspects result from the DPSIR analysis and the cross-check with international policy documents (UN, WHO). The identified aspects are categorized into nine sustainability characteristics, each consequently elaborated into five sustainability criteria. | This will be adjusted in the overall manuscript |
| | **Specific comments:** | Thank you for your detailed comments. | |
| 16 | - Line 40-44: delete (see general comments) | Will be deleted in abstract | See abstract |
| 17 | - Line 48: delete and give summary of results | Will be adjusted to: This resulted in the following set of hydrological, technical and socio-economic sustainability characteristics, respectively: (1) water quality, water resource availability, and impact of drinking water abstraction; (2) reliability and resilience of the technical system, and energy use and environmental impact; (3) drinking water availability, water governance, and land and water use. | See abstract |
| 18 | - Line 56-57: too general, delete | Will be deleted | Section 1 |
| 19 | - Line 57-60: give reference | Reference: WHO, & UNICEF. (2017). Progress on drinking water, sanitation and hygiene, 2017 update and SDG Baselines. | Section 1 |
| 20 | - Line 64-67: could be shortened (little information), only references sufficient | Will be adjusted to: For instance, two recent examples of water crises were reported in Cape Town, South Africa and São Paolo, Brazil (Sorensen, 2017, Cohen, 2016). | Section 1 |
| 21 | - Line 70-71: delete sentence | Will be deleted | Section 1 |
| 22 | - Line 84: delete sentence | Will be deleted | Section 1 |
| 23 | - Line 90-96: not much extra information (too general without references), so consider deleting. | Will be deleted | Section 2 |
| 24 | - Line 102-104: delete sentence | Will be deleted | Section 2 |
| 25 | - Line 127-130: not relevant information | Will be deleted | Section 2 |
| 26 | - Line 131: internal colleagues = staff | Will be adjusted | Section 2 |
| 27 | - Line 135: how the authors came to the defined "sustainability characteristics"? | The cases were analysed using the DPSIR framework. The sustainability aspects of these cases were identified in the descriptive results of the DPSIR analysis. The results were combined with Dutch governmental reports on these events and developments and cross-checked with Vitens staff. The sustainability aspects were categorized into hydrological, technical and socio-economic aspects. This resulted in a set of relevant sustainability aspects (in original manuscript in Table 1-3 and App A). The following step was used to broaden the perspective from the drinking water supply in the Netherlands to a more general perspective, by cross-checking the set of sustainability aspects with the targets and indicators in Sustainable Development Goal 6 (further referred to as "SDG 6", see App. D) (UN 2015), and the WHO Guidelines for Drinking-Water Quality (WHO 2017). Based on the analysis nine hydrological, technical and | This is elaborated in section 2 |

| | | | |
|---|---|---|---|
| | | socio-economic sustainability characteristics were proposed that cover the identified sustainability aspects. | |
| 28 | - Line 142: can be = could potentially be | Removed | Section 2 |
| 29 | - Line 144: Figure 1 does not give much extra information in relation to text so can be deleted. | We will make the method section more concise which gives the same information as the figure. Therefore we will remove this figure. | Figure is removed |
| 30 | - Line 147 and onwards: Section 2.1 is too general with a few references. Could be shortened to in or two sentences as introduction. | We will shorten and integrate section 2.1 into the introduction of section 2. | Section 2 |
| 31 | - Line 178 and onwards: could be more concised too, by at least deleting 178-182 | Sentences deleted and adjusted | Section 2 |
| 32 | - Line 210: Figure gives little extra information, so could be deleted. By the way, when it is not an own figure, a reference should be given. | We agree that the figure does not provide significant information additional to the method section, therefore we will remove the figure. | Figure is removed |
| 33 | - Line 214: case selection should be more to the point | We will adjust the section on the case selection to: 'In this research three drinking water supply cases in the Netherlands have been selected. The case studies were analysed to find sustainability aspects caused by the identified pressures and short- and/or long-term responses in each case, because short-term shocks have different impacts and call for other responses than long-term stresses (Smith and Stirling 2010). The cases therefore focus on short-term events as well as long-term developments All three cases also relate to targets set in SDG 6 (UN, 2015). The DPSIR analysis of the case studies is presented in Appendices A-C.' Additionally we will add the case descriptions to this section. | Section 2.2 |
| 34 | - Line 215-219: general information without references, could be deleted. | Will be deleted | Section 2.1 |
| 35 | - Line 233-240: avoid redundant information (already explained elsewhere) | Will be deleted | Section 2.1 |
| 36 | - Line 246 and onwards: use Italic for the DPSIR elements | Will be adjusted | Section 2.1 |
| 37 | - Line 246 and onwards: give references for the information that is given on the cases (e.g. line 247, 257, 259, 260-263, . . ..) - Line 260-263: how this information is obtained/verified? | The description of the cases is partially based on raw, unpublished operating data from Vitens, that are presented in Illustrations. 1, 2 and 3. The 2018 drought was evaluated by the Ministry of Infrastructure and Environment, which report was the main source of information. A reference to the unpublished Vitens data will be added to the sections. | Section 2.2 |
| 38 | - Line 274 and onwards: Is there a "case" or is it a "general" description. Now that is not clear.. Better, focus on the "Vitens case". | The 2nd case is focused on how the groundwater quality development affects the groundwater abstraction for drinking water supply in the Netherlands. The illustration is an example from Vitens practice, based on unpublished groundwater quality data. | Section 2.2 |
| 39 | - Line 343-352: redundant information, so delete. | The information will be integrated in the method section. | Section 2 |
| 40 | - Line 360: introduce JPM | This referred to the WHO Guidelines for drinking water quality and/or the WHO/UNICEF Joint Monitoring Programme for Water Supply, Sanitation and Hygiene. This will be adjusted into WHO Guidelines instead of using the abbreviation | Section 2, section 3. |

| 41 | - Line 367: check table 4, e.g. what is difference between "raw water" and "surface water" or "groundwater"; "impact of abstraction" is redundant; "water quantity" = "water flows". See also general comments. | Raw water = the water that is used for the drinking water production. This can be abstracted groundwater or surface water depending on the used water resource. Water resources availability refers to the availability of the water resources for drinking water production based on characteristics of the hydrological system, whereas the impact of the abstraction refers to the impact of the abstraction to the hydrological system, and depends of the size and nature of the abstraction. | Footnote on raw water

Appendix E |
|---|---|---|---|
| 42 | - Line 378-383: too speculative. Please stick to own findings (and discuss in relation to literature). | Will be adjusted | Section 3.1 |
| 43 | - Line 390: this will also impact costs of investments and thus water tariff. | Will be mentioned in section 3.3 | Section 3.3 |
| 44 | - Line 396: delete sentence | Will be deleted | Section 3.3 |
| 45 | - Line 444 and 469: why is the existence of a WSP a sustainability criterium? | Drinking water safety is a prerequisite for public health and sustainable drinking water supply. The WHO Guidelines consider water safety plans as essential to provide the basis for system protection and process control to ensure water quality issues present a negligible risk to public health and that water is acceptable to consumers. A WSP can be built on various safety protocols. We will add this to the manuscript and we will adjust the name of the criterion into 'water safety protocols'. | Section 3.3 |
| 46 | - Line 476 and onwards: avoid repetitions with previous sections, see earlier comments. | Table 7 will be adjusted in App E with an elaboration of the sustainability criteria. | Table is removed. Appendix E is added. |
| 47 | - Line 490- 501: delete (see general comments) | We will adjust this as a part of the discussion. | Section 4.2 |
| 48 | - Line 503-510: delete (see general comments) | Will be deleted | |
| 49 | - Line 521-529: delete (see general comments) | Will be deleted | |

**References**

- Binder, C. R., J. Hinkel, P. W. G. Bots and C. Pahl-Wostl (2013). "Comparison of frameworks for analyzing social-ecological systems." Ecology and Society **18**(4): 26-45.

- Carr, E. R., P. M. Wingard, S. C. Yorty, M. C. Thompson, N. K. Jensen and J. Roberson (2009). "Applying DPSIR to sustainable development." INT J SUST DEV WORLD **14**(6): 543-555.

- Ministry of Infrastructure and Environment and Ministry of Economic Affairs and Climate Policy (2019). Nederland beter weerbaar tegen droogte; Eindrapportage Beleidstafel Droogte (The Netherlands more resilient to drought; final report policy table drought). The Hague, the Netherlands**:** 77.

- Smith, A. and A. Stirling (2010). "The Politics of Social-ecological Resilience and Sustainable Socio-technical Transitions." ECOL SOC **15**(1): 11.

- UN (2015). Transforming our world: The 2030 Agenda for Sustainable Development, United Nations. **A/RES/70/1:** 21.

- UN (2018). Sustainable Development Goal 6 Synthesis Report 2018 on Water and Sanitation. U. Water. New York, USA, United Nations**:** 199.

- UNICEF and WHO (2015). Progress on Sanitation and Drinking Water; 2015 Update and MDG Assessment, UNICEF and World Health Organization**:** 90.

- Van Engelenburg, J., M. Fleuren and M. De Jonge (2020). Drinking water production data 2018-2020 and groundwater quality data 1985-2020 (unpublished). Vitens, Vitens.

- Van Noordwijk, M., E. Speelman, G. J. Hofstede, A. Farida, A. Y. Abdurrahim, A. Miccolis, A. L. Hakim, C. N. Wamucii, E. Lagneaux, F. Andreotti, G. Kimbowa, G. G. C. Assogba, L. Best, L. Tanika, M. Githinji, P. Rosero, R. R. Sari, U. Satnarain, S. Adiwibowo, A. Ligtenberg, C. Muthuri, M. Peña-Claros, E. Purwanto, P. Van Oel, D. Rozendaal, D. Suprayogo and A. J. Teuling (2020). "Sustainable Agroforestry Landscape Management: Changing the Game." Land **9**(8).

- Vitens (2016). Resiliently ahead; Long-term vision on our infrastructure 2016-2040. Zwolle, The Netherlands, Vitens**:** 36.

- WHO (2017). Guidelines for drinking-water quality: fourth edition incorporating the first addendum. Geneva, World Health Organization**:** 631.

- WHO and UNICEF (2017). Progress on drinking water, sanitation and hygiene, 2017 update and SDG Baselines. Geneva, World Health Organization (WHO) and UNICEF**:** 116.

---

## Author Response (AR1)

**Rebuttal**

| | |
|---|---|
| Journal Name: | Drinking Water Engineering and Science |
| Manuscript Number or original submission: | Dwes-2020-8 |
| Title of the original Manuscript: | Sustainability characteristics of drinking water supply |
| New Title of the Revised Manuscript | Sustainability characteristics of drinking water supply in the Netherlands |
| Type of the Article | Research article |
| Keywords old manuscript | Systems approach; DPSIR; drinking water supply; local scale; sustainability |
| New Keywords of the Revised Manuscript | Systems approach; drinking water supply; local scale; sustainability; the Netherlands |

*Dear editorial board,*

*Herewith we respond to the reviews of the anonymous referees. We hope this rebuttal will provide you a clear overview of our response and the adjustments we propose to make to the original manuscript to meet the referees' remarks.*

*Kind regards,*
*The authors*

| # | Referee #1 Comments () | Authors Comments | Adjustments in new manuscript |
|---|---|---|---|
| 1 | Comments: being a quantitative researcher with exposure to narrative style research, I could not appreciate the content presented. | This paper is part of an interdisciplinary research project on drinking water supply, performed by researchers both with quantitative as well as more qualitative disciplinary backgrounds (water resources management, hydrology, climate change studies, drinking water supply) aiming to contribute to the sustainability of drinking water supply. Additional to the more quantitative research on the hydrological impact of drinking water supply, the researchers also were confronted with the complexity of research on sustainable drinking water supply. This urged us to use a systems approach that allowed combining quantitative and qualitative characteristics. We propose to do this by identifying the most relevant challenges that must be addressed in policy development on sustainable drinking water | In the final paragraph section 1 we have added the following to the aim of the research: This research aims to propose a set of sustainability characteristics that describe the drinking water supply system on a local scale *to support policy- and decision-making on sustainable drinking water supply.* |

| # | Referee #1 Comments () | Authors Comments | Adjustments in new manuscript |
|---|---|---|---|
| | | supply, offering policy makers and planners an evidence-based approach for assessing sustainability of drinking water supply from their perspective. | _line 115-116_ |
| 2 | The authors allude to an integrated assessment based on system thinking for the first time but only stick to DPSIR framework without motivating its choice. There were mentions of socioecological and sociotechnical systems but I didnt see much content coming out from those respective disciplines, except perhaps DPSIR to certain extent. Why didnt the authors think of system dynamics models that explicitly incorporate feedbacks and are capable of integrating fast and slow dynamical systems. This also then extends to the way case studies were dealt with. Given that DPSIR approach is rather linear, I found key important aspects of feedbacks, synergies and tradeoffs between various driving, state, impact and response variables. For example, some pressures such as due to population growth might be influenced by policy responses of past actions such as providing reliable and abundant water supply. These are quite important if SDGs are to be investigated. In this regard I found the choice of the framework used by the authors as not well justified. | To reach the aim of this research to support policy development on sustainable drinking water supply, we chose to conduct three case studies to identify the main sustainability aspects in these cases. For this we decided to use DPSIR. DPSIR has previously been used for complex water systems by various well-known researchers in this field, such as Claudia Pahl-Wostl. In Binder, Hinkel et al. (2013) a comparison was made between various frameworks. The authors of that paper concluded that DPSIR was a policy framework that does not explicitly include development of a model, but aims at providing policy-relevant information, on pressures and responses on different scales. In Carr, Wingard et al. (2009) the use of DPSIR for sustainable development was evaluated. Although the authors were critical regarding the use of the DPSIR framework on national, regional or global scales, they considered application on a local scale appropriate. They concluded that practitioners can use DPSIR for local-scale studies because it assesses the place-specific nuances of multiple concerned stakeholders more realistically. In Van Noordwijk, Speelman et al. (2020) DPSIR was used to understand the joint multi-scale phenomena in the forest-water-people nexus and thus diagnosed issues to be addressed in serious games for local decision-making.

In the discussion we reflected on the limitations of the linear DPSIR approach with regard to the trade-offs and feedbacks in the drinking water supply. While the aim of the research was to identify sustainability characteristics for drinking water supply on a local scale to support policy development and stakeholder involvement rather than analysis and modeling of the system dynamics, we decided to use this framework. A next step could potentially be to use the identified system characteristics for a system dynamics analysis and modeling. However, this is beyond the scope of this current research. | This has been elaborated in the (new) section 2.1 on the case analysis method. Line 198-239.

In the new discussion section 4.1 this has been elaborated further (line 518-540). |
| 3 | I also had difficulties appreciating the discussion as I found tables synthesizing aspects of the three case studies repetitive. If the authors are intending to revise and resubmit, I would challenge the authors on providing a more sound basis for the choice of DPSIR framework in their pursuit of holistically assessing the sustainability of drinking water supply systems while not ignoring key aspects of feedbacks between | The tables of the case studies indeed show repetitive issues. This has been solved by removing the summarizing tables 1-3 and referring to the adjusted Appendices A-C. Concerning the remark on the dynamics in the sociohydrological, as well as the sociotechnical dynamics, we refer to the aim of the research, which was to identify the most relevant challenges that must be addressed in policy development on sustainable drinking water supply, rather than the system dynamics.
In our discussion we addressed the fact that the feedbacks and trade-offs in the drinking | Appendices A-C (adjusted/new)

See above.

Section 4.1 (line 518-540) has |

| # | Referee #1 Comments () | Authors Comments | Adjustments in new manuscript |
|---|---|---|---|
| | slow and fast dynamics of sociohydrological systems that supply systems are embedded in. | water supply cases complicated the DPSIR analysis. However, for the aim of the research, the DPSIR approach sufficed. Use of a different integrated systems approach would not have led to a significantly different outcome of the research. | been adjusted (discussion on use of DPSIR). |
| 4 | What could have been innovative would perhaps be a narrative treatment of how water supply systems might themselves have emerged from the underlying sociohydrological dynamics, locking them into a path towards unsustainable development (e.g. water supply systems that emerged in water abundant/flood prone countries might not be as resilient to drought events as those that emerged in latter drought prone systems). The case studies presented provide abundant material to shift the narrative in this direction. | Evaluation of how water supply systems developed as a result of underlying sociohydrological dynamics would indeed be a very interesting research topic. The case studies could definitely be used for this, when combined with case studies in semi-arid/arid countries. However, this is beyond the scope of the current study, which was to find sustainability characteristics. | No adjustment in manuscript. |

| # | Referee #2 Comments () | Author's comments | Adjustments in new manuscript |
|---|---|---|---|
| | Sustainability characteristics of water supply were determined based on the analysis of three case studies in the Netherlands. From there general sustainability criteria were identified that can be used in to assess drinking water supply. The paper tackles an interesting subject and is well written. However, it needs some adjustment before publication. | Thank you for your kind words. | |
| | **General comments:** | | |
| 5 | - The title should include the fact that the study was based on three case studies in the Netherlands. | Thank you for this suggestion. The title has been adjusted accordingly (adding "in The Netherlands"). | Title has been adjusted. |
| 6 | - The paper is rather long; the writing could be more concised and redundancies should be eliminated. | We have removed several figures, integrated some of the tables and removed redundancies following your suggestions. | Fig 1-2, Table 1-3 have been removed. |
| 7 | - When general statements are done, they should be supported by literature. | Where available references have been added. If not available, the statement has been removed. | e.g. 220-229 (Binder, Carr, Van Noordwijk) |
| 8 | - The methodology chapter is rather general, without a good description how sustainability characteristics and criteria were precisely determined. | The adopted research approach consisted of four steps. The first step was the selection and analysis of three drinking water practice cases in the Netherlands, aiming to identify the Dutch sustainability aspects in these cases. Three Dutch cases were selected based on their impact to the sustainability of drinking water supply in the Netherlands, illustrated with Vitens data (Van Engelenburg, Fleuren et al. 2020). In the second step the cases were analysed using the DPSIR framework (see section 2.1). | The method is more precisely described and clarified in section 2. Line 148-191. |

| | | The sustainability aspects of these cases were identified in the descriptive results of the DPSIR analysis. The results were combined with Dutch governmental reports on these events and developments (Vitens 2016, Ministry of Infrastructure and Environment and Ministry of Economic Affairs and Climate Policy 2019) and cross-checked with Vitens staff. The sustainability aspects were categorized into hydrological, technical and socio-economic aspects. This resulted in a set of relevant sustainability aspects. The following step was used to broaden the perspective from the drinking water supply in the Netherlands to a more general perspective, by cross-checking the set of sustainability aspects with the targets and indicators in Sustainable Development Goal 6 (UN 2015) and the WHO Guidelines for Drinking-Water Quality (WHO 2017). Based on the analysis nine hydrological, technical and socio-economic sustainability characteristics were proposed that cover the identified sustainability aspects.
In the final step of the study each sustainability characteristic was elaborated further into five sustainability criteria that describe the local drinking water supply system.
This resulted in a proposal for sustainability characteristics and criteria of local drinking water supply systems that could potentially be applied in various contexts. | |
|---|---|---|---|
| 9 | - In addition, the sustainability criteria should be better formulated in order to be able to judge compliance (or not). | We have provided an additional appendix E to the current paper that formulates and elaborates the sustainability criteria in the following information for each of the criteria: general explanation of the criterion, description of what may be considered sustainable, under pressure and unsustainable, and suggestions for indicators or other date sources. | Additional detailed information has been provided in Appendix E. |
| 10 | - When looking at the general use of the criteria for judgement of water supply systems in the world, at least some criteria are missing, such as Non-revenue water/leakage (which is maybe not a question in the Netherlands, but internationally it is) for technical system; and cost-recovery, a good-billing system, transparency in water tariffs, equality in water billing, etc. for governance. These flaws may be avoided by better (and more systematically) addressing previous bullets. | The mentioned criteria that the second referee found missing, are implicitly accounted for in the sustainability criteria. Non-revenue water/leakage in "Technical state distribution infrastructure", cost-recovery/billing system/tariffs are implicitly accounted for in the governance criteria "Availability of (drinking) water legislation and policies" and "Compliance of drinking water supplier". We have clarified this in the elaboration of the sustainability criteria in the appendix as mentioned above. | Additional detailed information has been provided in Appendix E. |
| 11 | - Description of cases should be part of methodology section. | The case description has been moved to the method section (section 2.2), and section 3 has been limited to the results of the analysis. | Section 2.2 and section 3 have been adjusted accordingly. |
| 12 | - Avoid repetition of results in the various tables. It is better to summarize at once and then describe in the various sections. | We integrated Table 1-3 into Appendix A-C. In addition, we have moved Table 7 towards Appendix E and elaborated this into a further description of the criteria. | App. A-C, App. E have been adjusted accordingly. |

| | | | |
|---|---|---|---|
| 13 | - Discussion (with literature) should be part of "results" section and not of "conclusions" and conclusions should be concised. | We have adjusted the final section, separating the discussion from the final conclusion of the paper in two separate sections, with specific attention to the conciseness of the writing. | Section 4 (Discussion) and Section 5 (Conclusion) have been adjusted accordingly. |
| 14 | - Language, including tenses, should be checked: present tense for general statements and past tense for own findings and work. | This has been checked and adjusted where applicable in the revised manuscript. | E.g. line 34, 37, 114, 120, 200, 261 |
| 15 | - Avoid word "issues", but better "characteristics" "criteria" "aspects", depending on own definition. | We have replaced the word 'sustainability issue' by 'sustainability aspect'. These aspects result from the DPSIR analysis and the cross-check with international policy documents (UN, WHO). The identified aspects are categorized into nine sustainability characteristics, each consequently elaborated into five sustainability criteria. | This has been adjusted where relevant throughout the manuscript. |
| | **Specific comments:** | Thank you for your detailed comments. | |
| 16 | - Line 40-44: delete (see general comments). | Has been deleted in abstract | See abstract. Line 39-44. |
| 17 | - Line 48: delete and give summary of results. | Has been adjusted into: This resulted in the following set of hydrological, technical and socio-economic sustainability characteristics: (1) water quality, water resource availability, and impact of drinking water abstraction; (2) reliability and resilience of the technical system, and energy use and environmental impact; (3) drinking water availability, water governance, and land and water use. | See abstract. Line 46-50. |
| 18 | - Line 56-57: too general, delete. | Has been deleted | Section 1. Line 62-63. |
| 19 | - Line 57-60: give reference. | Reference: WHO, & UNICEF. (2017). Progress on drinking water, sanitation and hygiene, 2017 update and SDG Baselines. | Section 1. Line 66. |
| 20 | - Line 64-67: could be shortened (little information), only references sufficient. | Has been adjusted to: For instance, two recent examples of water crises were reported in Cape Town, South Africa and São Paolo, Brazil (Sorensen, 2017, Cohen, 2016). | Section 1. Line 69-70. |
| 21 | - Line 70-71: delete sentence. | Has been deleted. | Section 1. Line 70-73. |
| 22 | - Line 84: delete sentence. | Has been deleted. | Section 1. Line 77-78. |
| 23 | - Line 90-96: not much extra information (too general without references), so consider deleting. | Has been deleted. | Section 1. Line 100-104. |
| 24 | - Line 102-104: delete sentence. | Has been deleted. | Section 2. Line 110-111. |
| 25 | - Line 127-130: not relevant information. | Has been deleted. | Section 2 has been rewritten. |
| 26 | - Line 131: internal colleagues = staff. | Has been adjusted. | Section 2. Line 173. |
| 27 | - Line 135: how the authors came to the defined "sustainability characteristics"? | The cases were analysed using the DPSIR framework. The sustainability aspects of these cases were identified in the descriptive results of the DPSIR analysis. The results were combined with Dutch governmental reports on these events and developments and cross-checked with Vitens staff. The sustainability aspects were categorized into hydrological, technical and socio-economic aspects. This resulted in a set of relevant sustainability aspects (in the original manuscript in Tables | This is elaborated in section 2 and 2.1. |

| | | 1-3 and App A). The following step was used to broaden the perspective from the drinking water supply in the Netherlands to a more general perspective, by cross-checking the set of sustainability aspects with the targets and indicators in Sustainable Development Goal 6 (further referred to as "SDG 6", see App. D) (UN 2015), and the WHO Guidelines for Drinking-Water Quality (WHO 2017). Based on the analysis nine hydrological, technical and socio-economic sustainability characteristics were proposed that cover the identified sustainability aspects. | |
|---|---|---|---|
| 28 | - Line 142: can be = could potentially be. | Original sentence has been removed. | Section 2 has been rewritten. |
| 29 | - Line 144: Figure 1 does not give much extra information in relation to text so can be deleted. | We have made the Methods section more concise. Because it gives the same information as the figure, we have removed this figure. | Figure has been removed. |
| 30 | - Line 147 and onwards: Section 2.1 is too general with a few references. Could be shortened to in or two sentences as introduction. | We have shortened and integrated section 2.1 of the original manuscript into the introduction of section 2. | Section 2 has been rewritten. |
| 31 | - Line 178 and onwards: could be more concised too, by at least deleting 178-182. | Sentences have been deleted or adjusted | Section 2 has been rewritten. |
| 32 | - Line 210: Figure gives little extra information, so could be deleted. By the way, when it is not an own figure, a reference should be given. | We agree that the figure does not provide significant information additional to the Methods section, therefore we have removed the figure. | Figure has been removed. |
| 33 | - Line 214: case selection should be more to the point. | We have adjusted the section on the case selection to: 'In this research three drinking water supply cases in the Netherlands have been selected. The case studies were analysed to find sustainability aspects caused by the identified pressures and short- and/or long-term responses in each case, because short-term shocks have different impacts and call for other responses than long-term stresses (Smith and Stirling 2010). The cases therefore focus on short-term events as well as long-term developments. All three cases also relate to targets set in SDG 6 (UN, 2015). The DPSIR analysis of the case studies is presented in Appendices A-C.'  Additionally, we will add the case descriptions to this section. | Section 2.2. Line 259-274.

Line 275 and further. |
| 34 | - Line 215-219: general information without references, could be deleted. | Has been deleted | Section 2.1 has been rewritten. |
| 35 | - Line 233-240: avoid redundant information (already explained elsewhere). | Has been deleted | Section 2.1 has been rewritten. |
| 36 | - Line 246 and onwards: use Italic for the DPSIR elements. | Has been adjusted | Section 2. Line 161 and further. |
| 37 | - Line 246 and onwards: give references for the information that is given on the cases (e.g. line 247, 257, 259, 260-263, . . ..)
- Line 260-263: how this information is obtained/verified? | The description of the cases is partially based on raw, unpublished operating data from Vitens, that are presented in Illustrations 1, 2 and 3. The 2018 drought was evaluated by the Ministry of Infrastructure and Environment, which report was the main source of information. A reference to the unpublished Vitens data (Van Engelenburg e.a., 2020) has been added. | Section 2.2, added to headings Figure 1-3. |
| 38 | - Line 274 and onwards: Is there a "case" or is it a "general" | The second case is focused on how the groundwater quality development affects the | Section 2.2. Line 230 e.v. |

| | | | |
|---|---|---|---|
| | description. Now that is not clear. Better, focus on the "Vitens case". | groundwater abstraction for drinking water supply in the Netherlands. The illustration is an example from Vitens practice, based on unpublished groundwater quality data. | |
| 39 | - Line 343-352: redundant information, so delete. | The redundant information has been removed, we have rewritten and adjusted the full Methods section (2/2.1/2.2). | Section 2 has been rewritten. |
| 40 | - Line 360: introduce JPM. | This referred to the WHO Guidelines for drinking water quality and/or the WHO/UNICEF Joint Monitoring Programme for Water Supply, Sanitation and Hygiene. This has been adjusted into WHO Guidelines instead of using the abbreviation. | Explained in section 2. Line 182. |
| 41 | - Line 367: check table 4, e.g. what is difference between "raw water" and "surface water" or "groundwater"; "impact of abstraction" is redundant; "water quantity" = "water flows". See also general comments. | Raw water = the water that is used for the drinking water production. This can be abstracted groundwater or surface water, depending on the used water resource. Water resources availability refers to the availability of the water resources for drinking water production based on characteristics of the hydrological system, whereas the impact of the abstraction refers to the impact of the abstraction to the hydrological system, and depends on the size and nature of the abstraction. | Footnote 1 on raw water. Line 335.(Appendix E). |
| 42 | - Line 378-383: too speculative. Please stick to own findings (and discuss in relation to literature). | Has been adjusted | Section 3.1. Line 408-410. |
| 43 | - Line 390: this will also impact costs of investments and thus water tariff. | Has been mentioned in section 3.3 | Section 3.3. Line 503-504. |
| 44 | - Line 396: delete sentence. | Has been deleted | Section 3.3. Line 426-427 |
| 45 | - Line 444 and 469: why is the existence of a WSP a sustainability criterium? | Drinking water safety is a prerequisite for public health and sustainable drinking water supply. The WHO Guidelines consider water safety plans as essential to provide the basis for system protection and process control to ensure water quality issues present a negligible risk to public health and that water is acceptable to consumers. A WSP can be built on various safety protocols. We have added this to the manuscript, and we adjusted the name of the criterion into 'water safety protocols'. | Section 3.3. Line 478-485. |
| 46 | - Line 476 and onwards: avoid repetitions with previous sections, see earlier comments. | Table 7 has been adjusted in App E with an elaboration of the sustainability criteria. | Table has been removed. Appendix E has been added. |
| 47 | - Line 490-501: delete (see general comments). | We have adjusted this as a part of the discussion. | Section 4.2 |
| 48 | - Line 503-510: delete (see general comments). | Has been deleted. | Line 541-551. |
| 49 | - Line 521-529: delete (see general comments). | Has been deleted. | Line 561-568. |

- # References

- Binder, C. R., J. Hinkel, P. W. G. Bots and C. Pahl-Wostl (2013). "Comparison of frameworks for analyzing social-ecological systems." Ecology and Society **18**(4): 26-45.

- Carr, E. R., P. M. Wingard, S. C. Yorty, M. C. Thompson, N. K. Jensen and J. Roberson (2009). "Applying DPSIR to sustainable development." INT J SUST DEV WORLD **14**(6): 543-555.

- Ministry of Infrastructure and Environment and Ministry of Economic Affairs and Climate Policy (2019). Nederland beter weerbaar tegen droogte; Eindrapportage Beleidstafel Droogte (The Netherlands more resilient to drought; final report policy table drought). The Hague, the Netherlands**: 77.

- Smith, A. and A. Stirling (2010). "The Politics of Social-ecological Resilience and Sustainable Socio-technical Transitions." ECOL SOC **15**(1): 11.

- UN (2015). Transforming our world: The 2030 Agenda for Sustainable Development, United Nations. **A/RES/70/1:** 21.

- UN (2018). Sustainable Development Goal 6 Synthesis Report 2018 on Water and Sanitation. U. Water. New York, USA, United Nations**: 199.

- UNICEF and WHO (2015). Progress on Sanitation and Drinking Water; 2015 Update and MDG Assessment, UNICEF and World Health Organization**: 90.

- Van Engelenburg, J., M. Fleuren and M. De Jonge (2020). Drinking water production data 2018-2020 and groundwater quality data 1985-2020 (unpublished). Vitens, Vitens.

- Van Noordwijk, M., E. Speelman, G. J. Hofstede, A. Farida, A. Y. Abdurrahim, A. Miccolis, A. L. Hakim, C. N. Wamucii, E. Lagneaux, F. Andreotti, G. Kimbowa, G. G. C. Assogba, L. Best, L. Tanika, M. Githinji, P. Rosero, R. R. Sari, U. Satnarain, S. Adiwibowo, A. Ligtenberg, C. Muthuri, M. Peña-Claros, E. Purwanto, P. Van Oel, D. Rozendaal, D. Suprayogo and A. J. Teuling (2020). "
[revised manuscript text omitted]

Sources of pollution
Contaminants
Emerging contaminants
Groundwater quality
Surface water quality
Raw water quality | Other water resources
Surface water quantity
Groundwater quantity
Vulnerability of the water system
Drought impact
Water discharge | Impact of abstraction
Groundwater levels
Abstraction volume
Balance between annual recharge and annual abstraction
Hydrological compensation |
| **SDG 6 targets[1]** | 6.3, 6.5 | 6.4, 6.5 | 6.4, 6.6 |
| **WHO Guidelines for Drinking-Water Quality** | Health risks from microbial contamination | Small- or large-scale emergencies caused by natural hazards, such as droughts, floods,  | - |

| Hydrological sustainability characteristics | Water quality | Water resource availability | Impact of drinking water abstraction |
|---|---|---|---|
| (WHO, 2017a) JMP[2] | Acceptability of the drinking water (salinization, hardness, colour) | earthquakes or forest fire | |
| Sustainability criteria | Current raw water quality
Chemical aspects of water quality
Microbial aspects of water quality
Acceptability aspects of water quality
Monitoring and evaluation of water quality trends | Surface water quantity
Groundwater quantity
Other available water resources
Vulnerability used water system  for contamination
Natural hazards and emergencies risk | Impact on surface water system
Impact on groundwater system
Balance between annual recharge and abstraction
Hydrological compensation
Spatial impact of abstraction facility/ storage/reservoir |

[1] SDG = Sustainable Development Goal; see App. V for summary of Sustainable Development Goal 6 targets and
indicators related to sustainability characteristics (UN, 2015)

*Water resource availability* for drinking water supply can be differentiated into the surface water and groundwater availability, as illustrated in Case 1 "2018 Summer drought". Other sustainability aspects are the vulnerability of the surface and/or groundwater system to the water quality being affected permanently by land use, as illustrated in the case

"Groundwater quality development".

The water resource availability can also be limited due to small- or large-scale emergencies caused by natural hazards, such as droughts, floods, earth-quakes or forest fires (WHO and UNICEF, 2017), that will put the sustainability of the local drinking water supply under pressure.

The *impact of the drinking water abstraction* to the hydrological system entails the impact to both the surface water system and the groundwater system, but also the balance between the annual drinking water abstraction volume and the annual recharge of the (local) water system. Whether the impact of the abstraction is or can possibly by compensated hydrologically is another sustainability aspect. The spatial impact of the local drinking water abstraction facility may also be a sustainability aspect: a drinking water facility requires a certain water storage area or reservoir, which might have a significant spatial impact in the area and thus might affect local stakeholders.

**3.2 Technical sustainability characteristics**

Three technical sustainability characteristics are proposed that summarise the technical aspects for the drinking water supply as found in the case studies: *reliability* and *resilience*

*of the technical infrastructure* and *energy use and environmental impact* of the drinking water supply (Table 2).

The *reliability* of the supply system is defined in this research as "the (un)likeliness of the technical system to fail" (Hashimoto et al., 1982). The current technical state of the drinking water production facility and the distribution infrastructure, and the complexity of the water treatment are important technical sustainability criteria for the local drinking water supply system. Other technical criteria that should be considered are the supply continuity of the facility, which stands for the capability to meet the set legal standards for drinking water supply under all circumstances, and the operational reliability, to solve technical failures without disturbance of the drinking water supply.

***Table  2*** *Summary of proposed technical sustainability characteristics, technical aspects from*
*case studies (see App. A-C), relevant SDG[1] indicators and WHO Guidelines for Drinking-*
*Water Quality (WHO, 2017a)[2] aspects, and technical sustainability criteria.*

| Technical sustainability characteristics | Reliability of technical infrastructure | Resilience of technical infrastructure | Energy use and environmental impact |
|---|---|---|---|
| **Sustainability aspects from case studies** | Drinking water pressure | Abstraction capacity | Energy use |
| | Drinking water treatment | Treatment capacity | Environmental impact |
| | Reliability of abstraction, treatment and distribution infrastructure | Treatment methods | Additional excipients |
| | | Distribution capacity | Wastewater |
| | | Resilience of technical infrastructure | Waste materials |
| **SDG 6 targets[1]** | 6.1, 6.4 | 6.1, 6.4 | 6.4 |

| | | | |
|---|---|---|---|
| **WHO Guidelines for Drinking-Water Quality (WHO, 2017a)**  | Safely managed drinking water services, i.e. improved drinking water source on premises, available when needed and free from contamination | Resilient technologies and processes

Upgrades of water treatment and storage capacity | Reliability of the energy supply
Renewability of the energy |
| **Sustainability criteria** | Technical state abstraction and treatment facility
Technical state distribution infrastructure
Complexity of water treatment
Supply continuity for customers
Operational reliability | Abstraction permit compared to annual drinking water demand
Production capacity compared to peak demand
Flexibility of treatment method
Technical innovations to improve resilience
Technical investments to improve resilience | Energy use of abstraction and treatment
Energy use of distribution
Environmental impact (additional excipients, wastewater, waste materials)
Reliability energy supply
Use of renewable energy |

[revised manuscript text omitted]

| Socio-economic sustainability characteristics | Drinking water availability | Water governance | Land and water use |
|---|---|---|---|
| **Sustainability** **aspects** **from case studies** | Customers
Drinking water availability
Drinking water demand
Drinking water tariff
Drinking water quality
Drinking water volume
Drinking water shortage
Emergencies, disturbances
Water saving | Abstraction permits
Drinking water standards
Water authorities
Water legislation, policy and regulations
Drinking water suppliers
Compliance
Stakeholders | Water use
Land use
Agriculture
Nature, groundwater-dependent ecosystems
Financial compensation

[revised manuscript text omitted]